# Noisy Dual Mirror Descent: A Near Optimal Algorithm for Jointly-DP Convex Resource Allocation

**Du Chen**[*]   **Geoffrey A. Chua**
Nanyang Business School, Nanyang Technological University, Singapore, 639798
chen1443@e.ntu.edu.sg, gbachua@ntu.edu.sg

## Abstract

We study convex resource allocation problems with $m$ hard constraints under $(\varepsilon, \delta)$-joint differential privacy (Joint-DP or JDP) in an offline setting. To approximately solve the problem, we propose a generic algorithm called Noisy Dual Mirror Descent. The algorithm applies noisy Mirror Descent to a dual problem from relaxing the hard constraints for private shadow prices, and then uses the shadow prices to coordinate allocations in the primal problem. Leveraging weak duality theory, we show that the optimality gap is upper bounded by $\mathcal{O}(\frac{\sqrt{m \ln (1/\delta)}}{\varepsilon})$, and constraint violation is no more than $\mathcal{O}(\frac{\sqrt{m \ln (1/\delta)}}{\varepsilon})$ per constraint. When strong duality holds, both preceding results can be improved to $\widetilde{\mathcal{O}}(\frac{\sqrt{\ln (1/\delta)}}{\varepsilon})$ by better utilizing the geometric structure of the dual space, which is neglected by existing works. To complement our results under strong duality, we derive a minimax lower bound $\Omega\left(\frac{m}{\varepsilon}\right)$ for any JDP algorithm outputting feasible allocations. The lower bound matches our upper bounds up to some logarithmic factors for $\varepsilon \geq \max\{1, 1/(n\gamma)\}$, where $n\gamma$ is the available resource level. Numerical studies further confirm the effectiveness of our algorithm.

## 1   Introduction

The resource allocation problem is a classic optimization problem and has many applications in machine learning, such as Internet advertising and personalized recommendation, among many others. The problem is typically modeled as a utility maximization problem, where the central decision maker has to properly allocate $m$ types of limited resources to $n$ agents in order to maximize the total utility of all agents. Each agent's data $z_i := (u_i(\cdot), \boldsymbol{a}_i(\cdot), \mathcal{X}_i)$ consists of three elements: (i) a utility function $u_i : \mathbb{R}_+^s \to \mathbb{R}_+$ that maps an allocation $\boldsymbol{x}_i$ to a utility scalar; (ii) a consumption function $\boldsymbol{a}_i : \mathbb{R}_+^s \to \mathbb{R}_+^m$ that maps an allocation $\boldsymbol{x}_i$ to a consumption vector; and (iii) a feasible set $\mathcal{X}_i \subseteq \mathbb{R}_+^s$ that may capture the agent's special requirements. With a dataset $\mathcal{D} := \{z_i\}_{i=1}^n$ of $n$ agents, the problem is modeled as

$$\text{(Resource Allocation Problem)} \quad \max_{\{\boldsymbol{x}_i\}_{i=1}^n} \quad \sum_{i=1}^n u_i(\boldsymbol{x}_i) \tag{1}$$

$$\text{s.t.} \quad \sum_{i=1}^n \boldsymbol{a}_i(\boldsymbol{x}_i) \leq n\gamma\boldsymbol{b}; \tag{2}$$

$$\boldsymbol{x}_i \in \mathcal{X}_i, \quad \forall i = 1, 2, \ldots, n, \tag{3}$$

where $n\gamma\boldsymbol{b} > \boldsymbol{0}$ is the available level of $m$ types of resources. Evidently, the goal in (1) is to find the best allocation $\boldsymbol{x}_i$ for each agent subject to resource-coupling constraints (2) and to a personal

---

[*]Corresponding author

38th Conference on Neural Information Processing Systems (NeurIPS 2024).

requirement constraint (3). When $u_i(\cdot)$ is concave, and $\boldsymbol{a}_i(\cdot)$, $\mathcal{X}_i$ are both convex, the problem reduces to a convex constrained problem that has been widely studied in optimization literature.

However, an emerging issue in allocation problems is data privacy. Because constraint (2) couples agents, the allocation decision to one agent would affect the allocation to another agent. When a group of agents collude, they can infer other agents' data by analyzing their received allocations. For example, when allocating limited budgets among education agencies to support students in poverty, a simple rule is to allocate budgets proportional to the number of students in poverty [SLWA22]. Agencies who notice a change in received funding may infer the financial status of students in other agencies. Realizing this potential leakage of its students' financial status, an agency may misreport the number of its students in poverty, compromising the original intention of the financial support.

To overcome privacy concerns in such cases, joint differential privacy (Joint-DP or JDP, for short), which is a relaxation of differential privacy (DP), has been adopted. It is found that JDP is more suitable than DP for the considered allocation problems [HHR$^+$14]. Essentially, JDP ensures agent $i$'s data cannot be accurately inferred by a group of collusive agents without $i$. It guarantees that the allocations received by the collusive agents are insensitive to agent $i$'s data. Considering that the allocation decision $\boldsymbol{x}_i$ is usually private to agent $i$ herself, the privacy guarantee by JDP is hence meaningful and sufficiently strong. The most handy way to achieve JDP is by the well-known Billboard Lemma [HHR$^+$14]: publish a DP public signal on a billboard accessible to all agents, and compute the allocation for agent $i$ based on the DP public signal and agent $i$'s own data only; then the final allocations $(\boldsymbol{x}_1, \ldots, \boldsymbol{x}_n)$ satisfy JDP. The Billboard Lemma has wide applications, ranging from convex optimization [HHR$^+$14, HHRW16, HZ18], to multi-armed bandits [SS18, HLL$^+$22], to reinforcement learning [VBKW20]. We also use it for privacy analysis. For clarification purposes, we highlight that, in problem (1), dataset $\{z_i\}_{i=1}^n := \{(u_i(\cdot), \boldsymbol{a}_i(\cdot), \mathcal{X}_i)\}_{i=1}^n$ is the private information to protect; resource level $n\gamma\boldsymbol{b}$ is treated as public information.

While important and fundamental, resource allocation problems under JDP are far from being well understood, in terms of both algorithm design and theoretical analysis. For algorithm design, most existing works adopt a "dual decomposition" idea [HHRW16, HZ18], which dualizes the coupling constraint (2) and iteratively seeks private minimizers of a dual problem to coordinate the allocation process in the primal problem. However, the primal-dual relationship and the geometric structure of the dual space are not well exploited, leaving substantial room for improvement. In terms of theoretical analysis, both privacy accounting and optimality analysis are short of modern standard. For privacy accounting, existing works [HHRW16, HZ18, HZ19] all use Advanced Composition, which suggests an unnecessarily large noise to be injected, significantly hindering algorithm practicability. For optimality analysis, [HHRW16] fails to take full advantage of strong assumptions made, and the analysis by [HZ18] is specific to linear packing problems. More importantly, there is no formal understanding of the fundamental trade-off between privacy and optimality in resource allocation problems under JDP, i.e. minimax lower bounds. Motivated by these gaps in the literature, we focus our study on the following three questions.

1. Can we design a generic algorithm for resource allocation problem (1), and provide better privacy and optimality analysis?

2. Can we better utilize the primal-dual relationship to improve performance?

3. Is our algorithm (near) optimal? What is the minimax lower bound?

**Our contributions**  First, we propose a generic algorithm Noisy Dual Mirror Descent, which follows a similar dual decomposition idea. We tighten the privacy analysis through Rényi DP, and analyze performance (both optimality and constraint violations) by better utilizing the primal-dual relationship. Second, for many cases where strong duality holds, we further improve performance by leveraging on the $\ell_1$ geometry of the dual space. Last, we complement previous analysis with a matching minimax lower bound for $\varepsilon \geq \max\{1, 1/(n\gamma)\}$, suggesting near optimality of our algorithm. We further conduct numerical experiments to show the effectiveness of our proposed algorithm. Table 1 summarizes our contributions and compares them with results in the literature.

**Our interpretation of algorithm performance**  Because the algorithms considered may output infeasible solutions, when assessing their performance, we should take into account both suboptimality in utility and constraint violations, i.e., the last two columns in Table 1. We therefore treat the sum of them as the ultimate performance. This idea admits a social welfare interpretation: when the central

Table 1: Comparison of various works on resource allocation problems under $(\varepsilon, \delta)$-JDP.

| | problem setup | | | | theoretical results | | |
|---|---|---|---|---|---|---|---|
| | $u_i(\cdot)$ | $\boldsymbol{a}_i(\cdot)$ | $\mathcal{X}_i$ | assume strong duality? | LB | utility loss UB | total constraint violation |
| [HHRW16][‡] | linear | linear | probability simplex | ✓ | - | $\widetilde{\mathcal{O}}\left(\frac{m\sqrt{\ln(1/\delta)}}{\varepsilon}\right)$ (Theorem 3.3, w.h.p.) | $\widetilde{\mathcal{O}}\left(\frac{m\sqrt{\ln(1/\delta)}}{\varepsilon}\right)$ (Theorem 3.3, w.h.p.) |
| [HZ18][§] | linear | linear | $[\boldsymbol{0},\boldsymbol{1}]$ | ✓ | - | $\widetilde{\mathcal{O}}\left(\frac{\sqrt{m\ln(1/\delta)}}{\varepsilon}\right)$ (Theorem 1.1, w.h.p.) | $\widetilde{\mathcal{O}}\left(\frac{m^{3/2}\sqrt{\ln(1/\delta)}}{\varepsilon}\right)$ (section 3.2.2, w.h.p.) |
| This work | concave | convex | convex | ✗ | - | $\mathcal{O}\left(\frac{\sqrt{m\ln(1/\delta)}}{\varepsilon}\right)$ (Theorem 3.5) | $\mathcal{O}\left(\frac{m^{3/2}\sqrt{\ln(1/\delta)}}{\varepsilon}\right)$ (Theorem 3.6) |
| | | | | ✓ | $\Omega\left(\frac{m}{\varepsilon}\right)$[†] (Theorem 4.3) | $\widetilde{\mathcal{O}}\left(\frac{\sqrt{\ln(1/\delta)}}{\varepsilon}\right)$ (Theorem 3.10) | $\widetilde{\mathcal{O}}\left(\frac{m\sqrt{\ln(1/\delta)}}{\varepsilon}\right)$ (Theorem 3.10) |

*Notes.* All results in the table are stated for $\varepsilon \leq \ln(1/\delta)$ for conciseness; $m, n, T$ are the number of constraints, agents, and iterations, respectively. Tilde symbol $\widetilde{\mathcal{O}}$ hides poly-log factors in $m$; for results in [HHRW16, HZ18], $\widetilde{\mathcal{O}}$ additionally hides poly-log terms in $n$ and $T$. LB=lower bound, UB=upper bound, w.h.p=with high probability. [‡] The algorithm in [HHRW16] can be practically applied to general convex problems. But one supporting Lemma (Theorem 2.1) they used to derive analytical results is only valid for linear problems with solutions in probability simplex. [§] An lower bound $\widetilde{\Omega}(\frac{\sqrt{m}}{\varepsilon})$ can be derived from [HZ18, Theorem 1.2]. But their original statement is for the minimal supply, not for suboptimality as we considered. [†] The lower bound is for algorithms outputting feasible allocations, while upper bounds are achieved by algorithms that may output infeasible allocations. So, the lower bound could be higher. The lower bound holds only for $\varepsilon \geq \max\{1, 1/(n\gamma)\}$ where $\gamma \in (0, 1)$.

decision maker (e.g., a government) desires to implement an infeasible allocation, she may purchase additional resources from an emergency supplier to make the allocation feasible. Then, the total loss in social welfare is the sum of (i) the loss in utility of agents and (ii) the decision maker's expenditure on extra resources. If we only look at suboptimality, we may mistakenly conclude that the gap could be arbitrarily small. Following our interpretation of performance, our proposed algorithm is near optimal.

**Related work** JDP was initially proposed by [KPRU14] as a relaxation of differential privacy [DR14], which better fits the nature of privacy issues in zero-sum games, such as allocation problems [HHR+14, CKRW15]. Following this stream, the most relevant works to ours are [HHRW16, HZ18], both of which designed their algorithms with a dual decomposition idea. [HHRW16] proposed the first generic method, dual gradient descent, for solving allocation problems. For the special case of linear packing problem, [HZ18] designed a dual multiplicative weight update algorithm, and extended it to online setting. [GU22] further provides an economic interpretation of payoff sharing. Our work is also closely related to Mirror Descent (MD) [NY83], a generalization of projected gradient descent to non-Euclidean settings. Its private version, Noisy MD, has recently found many applications in non-Euclidean DP stochastic convex optimization [AFKT21, BGN21] and saddle-point problems [GGP24]. We further apply Noisy MD to JDP resource allocation problems.

## 2 Preliminaries

**Definition 2.1** (Differential privacy, [DR14]). *A mechanism $\mathcal{M}: \mathcal{Z}^n \to \mathcal{P}$ is $(\varepsilon, \delta)$-differentially private if, for any pair of neighboring datasets $\mathcal{D} \sim \mathcal{D}'$ that differ in one data point, and for any subset of output $\mathcal{S} \subseteq \mathcal{P}$, we have $\mathbb{P}[\mathcal{M}(\mathcal{D}) \in \mathcal{S}] \leq e^\varepsilon \cdot \mathbb{P}[\mathcal{M}(\mathcal{D}') \in \mathcal{S}] + \delta$.*

Throughout the work, we use the subscript $_{-i}$ to indicates variables without agent $i$. For example, $\mathcal{X}_{-i} := \mathcal{X}_1 \times \cdots \mathcal{X}_{i-1} \times \mathcal{X}_{i+1} \cdots \times \mathcal{X}_n$ is a feasible space without $i$, and $\mathcal{M}(\mathcal{D})_{-i} := (\mathcal{M}(\mathcal{D})_1, \ldots, \mathcal{M}(\mathcal{D})_{i-1}, \mathcal{M}(\mathcal{D})_{i+1}, \ldots, \mathcal{M}(\mathcal{D})_n)$ is the view from other agents without $i$.

**Definition 2.2** (Joint differential privacy, [KPRU14]). *A mechanism $\mathcal{M}: \mathcal{Z}^n \to \mathcal{X}^n$ is $(\varepsilon, \delta)$-jointly differentially private if for any pair of neighboring datasets $\mathcal{D} \sim \mathcal{D}'$ that differ in datapoint $i \in [n]$, and for any subset of outputs $\mathcal{S} \subseteq \mathcal{X}_{-i}$, we have $\mathbb{P}[\mathcal{M}(\mathcal{D})_{-i} \in \mathcal{S}] \leq e^\varepsilon \cdot \mathbb{P}[\mathcal{M}(\mathcal{D}')_{-i} \in \mathcal{S}] + \delta$.*

**Lemma 2.3** (Billboard Lemma, [HHR+14]). *Suppose $\mathcal{M} : \mathcal{Z}^n \to \mathcal{P}$ is $(\varepsilon, \delta)$-DP. For any given function $f : \mathcal{Z} \times \mathcal{P} \to \mathcal{X}$, denote the output of $f$ on individual $i$'s data as $f_i := f(z_i, \mathcal{M}(\mathcal{D}))$. Then, $(f_1, \ldots, f_n)$ is $(\varepsilon, \delta)$-JDP.*

**Definition 2.4** ($\alpha$-strong convexity). *Let $\alpha > 0$. Function $\ell : \mathbb{R}^m \to \mathbb{R}$ is said to be $\alpha$-strongly convex w.r.t. $\|\cdot\|_p$ over set $\mathcal{W}$, if $\ell(\boldsymbol{x}) \geq \ell(\boldsymbol{y}) + \langle \nabla \ell(\boldsymbol{y}), \boldsymbol{x} - \boldsymbol{y} \rangle + \frac{\alpha}{2} \|\boldsymbol{x} - \boldsymbol{y}\|_p^2, \forall \boldsymbol{x}, \boldsymbol{y} \in \mathcal{W}$.*

**Primal problem in a condensed form**    Let $\boldsymbol{x} := (\boldsymbol{x}_1, \ldots, \boldsymbol{x}_n)$ be the collection of allocations. For a given dataset $\mathcal{D}$, let $\mathsf{F}(\boldsymbol{x}) := \sum_{i=1}^n u_i(\boldsymbol{x}_i)$ be the total utility when allocation is $\boldsymbol{x}$. Let $\boldsymbol{a}(\boldsymbol{x}) := \sum_{i=1}^n \boldsymbol{a}_i(\boldsymbol{x}_i)$ be the total consumed resource, and let $\mathcal{X} := \mathcal{X}_1 \times \cdots \times \mathcal{X}_n$ be the feasible region. Then, the resource allocation problem (1), referred to as "primal problem" later, can be written in a condensed form:

$$\text{(Primal Problem)} \qquad \max_{\boldsymbol{x} \in \mathcal{X}} \{\mathsf{F}(\boldsymbol{x}) : \boldsymbol{a}(\boldsymbol{x}) \leq n\gamma\boldsymbol{b}\}. \tag{4}$$

The optimal allocation is denoted as $\boldsymbol{x}^* := \arg\max_{\boldsymbol{x} \in \mathcal{X}} \{\mathsf{F}(\boldsymbol{x}) : \boldsymbol{a}(\boldsymbol{x}) \leq n\gamma\boldsymbol{b}\}$.

**Dual problem**    By dualizing the coupling constraint with shadow prices $\boldsymbol{p} \geq \boldsymbol{0}$, we get a Lagrangian function $\min_{\boldsymbol{p} \geq \boldsymbol{0}} \max_{\boldsymbol{x} \in \mathcal{X}} \{\mathsf{F}(\boldsymbol{x}) + \langle \boldsymbol{p}, n\gamma\boldsymbol{b} - \boldsymbol{a}(\boldsymbol{x}) \rangle\}$. Thus, the Lagrangian dual problem is

$$\text{(Dual Problem)} \qquad \min_{\boldsymbol{p} \geq \boldsymbol{0}} \mathsf{D}(\boldsymbol{p}), \tag{5}$$

where $\mathsf{D}(\boldsymbol{p}) := \max_{\boldsymbol{x} \in \mathcal{X}} \{\mathsf{F}(\boldsymbol{x}) + \langle \boldsymbol{p}, n\gamma\boldsymbol{b} - \boldsymbol{a}(\boldsymbol{x}) \rangle\}$. It is easy to see that, given $\boldsymbol{p} \geq \boldsymbol{0}$, the dual problem is decomposable across agents, i.e., $\mathsf{D}(\boldsymbol{p}) = \sum_{i=1}^n \max_{\boldsymbol{x}_i \in \mathcal{X}_i} \{u_i(\boldsymbol{x}) + \langle \boldsymbol{p}, \gamma\boldsymbol{b} - \boldsymbol{a}_i(\boldsymbol{x}_i) \rangle\}$. Let us denote the dual problem's minimizer by $\boldsymbol{p}^* := \arg\min_{\boldsymbol{p} \geq \boldsymbol{0}} \mathsf{D}(\boldsymbol{p})$.

When there are multiple $\boldsymbol{x}^*$ and $\boldsymbol{p}^*$, we can break ties arbitrarily. Let $\|\cdot\|_p$ denote the $p$-norm of a vector, and let $[n] := \{1, \ldots, n\}$ be a running set. We impose some assumptions on dataset $\mathcal{D}$ to make both primal and dual problems interesting.

**Assumption 2.5** (Interesting instances). *For a dataset $\mathcal{D} := \{(u_i(\cdot), \boldsymbol{a}_i(\cdot), \mathcal{X}_i)\}_{i=1}^n$, we assume*

1. **Convexity**: *for each $i \in [n]$, utility function $u_i$ is concave on $\mathcal{X}_i$, consumption function $\boldsymbol{a}_i$ is convex on $\mathcal{X}_i$, and $\mathcal{X}_i$ is a convex set;*

2. **Boundedness**: *$\exists \bar{u} > 0$ and bounded $\boldsymbol{b} > \boldsymbol{0}$ with $\|\boldsymbol{b}\|_2 \leq B$ such that $u_i(\boldsymbol{x}_i) \in [0, \bar{u}]$ and $\boldsymbol{a}_i(\boldsymbol{x}_i) \in [\boldsymbol{0}, \boldsymbol{b}], \forall \boldsymbol{x}_i \in \mathcal{X}_i, i \in [n]$. Specially, if $\boldsymbol{0} \in \mathcal{X}_i$, then $u_i(\boldsymbol{0}) = 0$ and $\boldsymbol{a}_i(\boldsymbol{0}) = \boldsymbol{0}$;*

3. **Limited resource**: *the constant $\gamma$ in (4) is assumed to be in the interval $(0, 1)$. The primal problem (4) is feasible, and optimal $\boldsymbol{x}^*$ is attainable. Under $\boldsymbol{x}^*$, at least one of $m$ constraints in (4) is binding. Moreover, the optimal shadow prices $\boldsymbol{p}^*$ to dual problem (5) are not all zeros, i.e., $\|\boldsymbol{p}^*\|_1 \neq 0$;*

4. **Compulsory request allowed**: *request $i$ can be compulsory in the sense that $\mathcal{X}_i \not\ni \boldsymbol{0}$, only if $\max_{\boldsymbol{x}_i \in \mathcal{X}_i} \{u_i(\boldsymbol{x}_i) + \langle \boldsymbol{p}^*, -\boldsymbol{a}_i(\boldsymbol{x}_i) \rangle\} \geq 0$.*

These assumptions are very mild. The first assumption restricts our attention to convex problems. The second one assumes both utility and consumption functions are non-negative and bounded from above, and are both zeros if no resource is allocated. The third one assumes the non-private problem is indeed resource-constrained and not ill-posed. The constant $\gamma$ controls available resource levels, and $n\gamma$ means the number of agents whose requests can be fully fulfilled. In practice, usually $\gamma \geq 1/n$. The fourth one allows agents to propose compulsory requests with minimal requirements strictly greater than $\boldsymbol{0}$, as long as allocating resource to the agent is beneficial. Our assumptions are weaker than those in the literature [HZ18, BLM20, BLM23] by allowing compulsory requests.

## 3    The algorithm and analysis

Our algorithm is a private version of Mirror Descent applied to the dual problem (5). Mirror Descent (MD) [NY83] is a generalization of Projected Gradient Descent (Proj-GD) that can better cater to the geometry of the problem at hand by the proper choice of a strongly convex potential function $\Phi : \mathcal{W} \to \mathbb{R}^+$. The potential function chosen should meet some conditions below.

**Condition 3.1** (Potential Functions). *The potential function $\Phi : \mathcal{W} \to \mathbb{R}_+$ chosen should be (i) differentiable on $\text{int}(\mathcal{W})$, i.e., the interior of its domain $\mathcal{W} \subseteq \mathbb{R}^m$; and (ii) $\alpha$-strongly convex with respect to $\|\cdot\|_p$ on $\mathcal{W}$. For a given $\Phi$, let $B_\Phi(\boldsymbol{x}, \boldsymbol{y}) := \Phi(\boldsymbol{x}) - \Phi(\boldsymbol{y}) - \langle \nabla\Phi(\boldsymbol{y}), \boldsymbol{x} - \boldsymbol{y} \rangle$ be the Bregman divergence between $\boldsymbol{x} \in \mathcal{W}, \boldsymbol{y} \in \text{int}(\mathcal{W})$.*

With the primal problem (4) and dual problem (5) in mind, and a well-chosen potential function, we are ready to present the algorithm, Noisy Dual Mirror Descent. The algorithm is an iterative

---

**Algorithm 1** Noisy Dual Mirror Descent for Resource Allocation Problems, $\mathcal{A}$

---

**Parameters:** privacy parameters $(\varepsilon, \delta)$, variance $\sigma^2$; stepsizes $\{\eta^{(t)}\}_{t=1}^T$; potential function $\Phi(\cdot) : \mathcal{W} \to \mathbb{R}_+$ that is $\alpha$-strongly convex with respect to $\ell_p$-norm; conjugate index $q$ s.t. $1/q + 1/p = 1$, number of iterations $T$, feasible region of shadow prices $\mathcal{P} := \mathbb{R}_+^m$.

1: Set initial point $\boldsymbol{p}^{(1)} \in \mathcal{P} \cap \text{int}(\mathcal{W})$
2: **for** $t = 1$ to $T$ **do**
3:     Get intermediate allocation decision $\boldsymbol{x}^{(t)} \leftarrow \arg\max_{\boldsymbol{x} \in \mathcal{X}} \mathsf{F}(\boldsymbol{x}) + \langle \boldsymbol{p}^{(t)}, \gamma n \boldsymbol{b} - \boldsymbol{a}(\boldsymbol{x}) \rangle$
4:     Draw a noise vector $\mathbf{n}^{(t)} \sim \mathcal{N}(\mathbf{0}, \sigma^2 \boldsymbol{I}_{m \times m})$
5:     Update private shadow prices according to Noisy Mirror Descent:

$$\boldsymbol{p}^{(t+1)} \leftarrow \arg\min_{\boldsymbol{p} \in \mathcal{P} \cap \mathcal{W}} \left\{ \eta^{(t)} \cdot \left\langle \mathbf{g}^{(t)} + \mathbf{n}^{(t)}, \boldsymbol{p} \right\rangle + B_\Phi(\boldsymbol{p}, \boldsymbol{p}^{(t)}) \right\}, \tag{6}$$

    where $\mathbf{g}^{(t)} := n\gamma \boldsymbol{b} - \boldsymbol{a}(\boldsymbol{x}^{(t)})$
6: Final allocation decision $\boldsymbol{x}^{\mathcal{A}} := (\boldsymbol{x}_1^{\mathcal{A}}, \dots, \boldsymbol{x}_n^{\mathcal{A}}) \leftarrow \frac{1}{T} \sum_{t=1}^T \boldsymbol{x}^{(t)}$
7: **Output** $\boldsymbol{x}_i^{\mathcal{A}}$ to agent $i$, for all $i \in [n]$

---

algorithm, and in each iteration, we first calculate allocation decisions based on current shadow prices $\boldsymbol{p}^{(t)}$, i.e. Step 3. Then, we update the shadow prices by Noisy Mirror Descent update rule (6). The gradient $\mathbf{g}^{(t)}$ used is exactly the gradient of the dual problem $\mathsf{D}(\boldsymbol{p}^{(t)})$ according to Danskin's theorem. After $T$ iterations, we output the averaging allocations across all iterations. In other words, we basically apply Noisy Mirror Descent to solve the dual problem, and use the sequence of shadow prices to coordinate allocations in the primal problem. Intuitively, the shadow prices are implicitly pricing each resource to ensure that limited resources are allocated to agents most in need.

Recall the Billboard Lemma, if we can privatize the entire sequence of shadow prices in a DP manner, then the final allocation decisions will be JDP. Compared to existing works invoking Advanced Composition, we provide a tighter privacy accounting through Rényi DP.

**Theorem 3.2** (JDP Guarantee). *Given $\varepsilon > 0, \delta \in (0, 1)$ and $T \geq 1$, if noise variance $\sigma^2 = T \cdot c_{\varepsilon,\delta}$ with $c_{\varepsilon,\delta} := \|\boldsymbol{b}\|_2^2 \cdot \left( \frac{2\ln(1/\delta)}{\varepsilon^2} + \frac{1}{\varepsilon} \right)$, then Algorithm 1 $\mathcal{A}$ is $(\varepsilon, \delta)$-JDP.*

To better align with expressions in DP literature, we can assume $\varepsilon \leq \ln(1/\delta)$. Then, the magnitude of $c_{\varepsilon,\delta}$ becomes $\frac{\ln(1/\delta)}{\varepsilon^2}$, a magnitude of Gaussian Mechanism that frequently appears in DP literature. Moreover, the privacy analysis via Rényi DP significantly lowers the variance level. For example, when $\varepsilon = 1$, $\delta = 10^{-3}$, and $\|\boldsymbol{b}\|_2^2 = 1$, the variance indicated by Theorem 3.2 is $14.8T$, compared to approximately $121.6T$ by [HHRW16, Theorem 3.2], [HZ18, Lemma 3.1].

## 3.1 Performance upper bounds

We now move on to analyze the utility optimality gap $\mathsf{F}(\boldsymbol{x}^*) - \mathbb{E}_\mathcal{A}\left[\mathsf{F}(\boldsymbol{x}^{\mathcal{A}})\right]$. The following weak duality lemma bridges the primal and dual problems.

**Lemma 3.3** (Weak duality). *For any $\boldsymbol{p} \geq \mathbf{0}$, the objective value of dual problem (5) is always greater than or equal to that of primal problem (4), i.e., $\mathsf{D}(\boldsymbol{p}) \geq \max_{\boldsymbol{x} \in \mathcal{X}} \{\mathsf{F}(\boldsymbol{x}) : \boldsymbol{a}(\boldsymbol{x}) \leq n\gamma\boldsymbol{b}\}, \forall \boldsymbol{p} \geq \mathbf{0}$.*

An immediate result of weak duality is $\mathsf{D}(\boldsymbol{p}^{(t)}) \geq \mathsf{F}(\boldsymbol{x}^*), \forall t$. Hence, applying weak duality and Jensen's inequality, the optimality gap $\mathsf{F}(\boldsymbol{x}^*) - \mathsf{F}(\boldsymbol{x}^{\mathcal{A}})$ can be upper bounded as follows, provided

the randomness dice of $\mathcal{A}$ is fixed,

$$\mathsf{F}(\boldsymbol{x}^*) - \mathsf{F}(\boldsymbol{x}^{\mathcal{A}}) \leq \frac{1}{T}\sum_{t=1}^{T}\left(\mathsf{D}(\boldsymbol{p}^{(t)}) - \mathsf{F}(\boldsymbol{x}^{(t)})\right) = \frac{1}{T}\sum_{t=1}^{T}\left\langle\boldsymbol{p}^{(t)}, n\gamma\boldsymbol{b} - \boldsymbol{a}(\boldsymbol{x}^{(t)})\right\rangle$$

$$= \frac{1}{T}\sum_{t=1}^{T}\left\langle\boldsymbol{p}^{(t)}, \mathbf{g}^{(t)} + \mathbf{n}^{(t)}\right\rangle - \frac{1}{T}\sum_{t=1}^{T}\left\langle\boldsymbol{p}^{(t)}, \mathbf{n}^{(t)}\right\rangle,$$

where the first equality is by definition of the dual problem, and the second equality is by definition of gradient $\mathbf{g}^{(t)}$. Taking expectation with respect to $\mathcal{A}$ removes the subtrahend, since $\mathbf{n}^{(t)}$ is independent of $\boldsymbol{p}^{(t)}$ and zero-mean, which gives $\mathsf{F}(\boldsymbol{x}^*) - \mathbb{E}_{\mathcal{A}}\left[\mathsf{F}(\boldsymbol{x}^{\mathcal{A}})\right] \leq \mathbb{E}_{\mathcal{A}}\left[\frac{1}{T}\sum_{t=1}^{T}\left\langle\boldsymbol{p}^{(t)}, \mathbf{g}^{(t)} + \mathbf{n}^{(t)}\right\rangle\right]$. The inner term is closely related to stationarity gap [GL13, ABG+23] in nonconvex optimization. It is well-known that Proj-GD guarantees a small stationarity gap in both non-private [JNG+21] and private [ABG+23] cases. Not surprisingly, MD can also achieve a small stationarity gap.

**Lemma 3.4** (Cumulative stationarity gap of MD). *Suppose stepsizes $\eta^{(t)} = \eta$ in MD update rule (6) are the same for all $t \in [T]$, and let $\{\widetilde{\mathbf{g}}^{(t)}\}_{t=1}^{T}$ be gradients used for update. Then the cumulative stationarity gap $\sum_{t=1}^{T}\left\langle\boldsymbol{p}^{(t)} - \boldsymbol{p}, \widetilde{\mathbf{g}}^{(t)}\right\rangle$ for any anchor point $\boldsymbol{p} \in \mathcal{P} \cap \mathcal{W}$ is upper bounded as*

$$\sum_{t=1}^{T}\left\langle\boldsymbol{p}^{(t)} - \boldsymbol{p}, \widetilde{\mathbf{g}}^{(t)}\right\rangle \leq \frac{\eta\sum_{t=1}^{T}\left\|\widetilde{\mathbf{g}}^{(t)}\right\|_q^2}{2\alpha} + \frac{B_{\Phi}(\boldsymbol{p}, \boldsymbol{p}^{(1)})}{\eta}, \quad \forall\boldsymbol{p} \in \mathcal{P} \cap \mathcal{W}.$$

With Lemma 3.4 and proper stepsizes, we can control the optimality gap. Denote $\bar{\gamma} := \max\{\gamma, 1-\gamma\}$, $G := \bar{\gamma}^2 n^2 \|\boldsymbol{b}\|_q^2$, and let $\boldsymbol{z} \sim \mathcal{N}(\mathbf{0}, \boldsymbol{I}_{m\times m})$ be a standard Gaussian random vector. For a given potential function $\Phi$ and an initial point $\boldsymbol{p}^{(1)}$, denote $C_{\Phi}(\boldsymbol{p}^{(1)}) := \sqrt{B_{\Phi}(\mathbf{0}, \boldsymbol{p}^{(1)})/\alpha}$.

**Theorem 3.5** (Utility guarantee). *Set stepsizes $\eta^{(t)} = \sqrt{\frac{\alpha B_{\Phi}(\mathbf{0}, \boldsymbol{p}^{(1)})}{T\cdot(G+\sigma^2\mathbb{E}[\|\boldsymbol{z}\|_q^2])}}, \forall t \in [T]$. Suppose potential function's domain includes $\mathbf{0}$, i.e., $\mathbf{0} \in \mathcal{W}$. Then running algorithm $\mathcal{A}$ with iterations $T \geq \frac{G}{c_{\varepsilon,\delta}\cdot\mathbb{E}[\|\boldsymbol{z}\|_q^2]}$ and $\sigma^2$ chosen in Theorem 3.2 yields*

$$\mathsf{F}(\boldsymbol{x}^*) - \mathbb{E}_{\mathcal{A}}\left[\mathsf{F}(\boldsymbol{x}^{\mathcal{A}})\right] \leq 4C_{\Phi}(\boldsymbol{p}^{(1)})\sqrt{\mathbb{E}\left[\|\boldsymbol{z}\|_q^2\right]} \cdot \sqrt{c_{\varepsilon,\delta}}. \tag{7}$$

An instantiation of the utility guarantee is by $\Phi(\cdot) = \frac{1}{2}\|\cdot\|_2^2$, which is 1-strongly convex w.r.t. $\ell_2$-norm on $\mathbb{R}^m$. Then, the upper bound in (7) becomes $2\sqrt{2m}\left\|\boldsymbol{p}^{(1)}\right\|_2 \cdot \sqrt{c_{\varepsilon,\delta}}$, which depends on the choice of initial point $\boldsymbol{p}^{(1)}$. It seems the bound can be arbitrarily small if $\boldsymbol{p}^{(1)}$ is close to $\mathbf{0}$. But in fact, the theorem itself does not reflect the whole picture, since $\boldsymbol{x}^{\mathcal{A}}$ could be infeasible. Therefore, we have to further examine $\mathcal{A}$'s feasibility guarantee.

**Theorem 3.6** (Feasibility guarantee). *Given allocation decision $\boldsymbol{x}^{\mathcal{A}}$, denote the violation levels as $\boldsymbol{v}^{\mathcal{A}} := (\boldsymbol{a}(\boldsymbol{x}^{\mathcal{A}}) - n\gamma\boldsymbol{b})^+$, where positive part operator $(\cdot)^+$ applies element-wisely. In addition to Condition 3.1, additionally assume the domain $\mathcal{W}$ of potential function $\Phi$ contains $2\|\boldsymbol{p}^*\|_1\boldsymbol{e}_j, \forall j \in [m]$. Then, running algorithm $\mathcal{A}$ with the same setting as in Theorem 3.5 yields*

$$\mathbb{E}_{\mathcal{A}}\left[\left\|\boldsymbol{v}^{\mathcal{A}}\right\|_{\infty}\right] \leq \left(\frac{2\sqrt{\mathbb{E}\left[\|\boldsymbol{z}\|_q^2\right]} \cdot C_{\Phi,1}(\boldsymbol{p}^{(1)})}{\sqrt{\alpha} \cdot \|\boldsymbol{p}^*\|_1} + 2\sqrt{2\ln m}\right) \cdot \sqrt{c_{\varepsilon,\delta}}, \tag{8}$$

*where $C_{\Phi,1}(\boldsymbol{p}^{(1)}) = \frac{B_{\Phi}(\mathbf{0}, \boldsymbol{p}^{(1)}) + \max_{\boldsymbol{e}\in E_1} B_{\Phi}(2\|\boldsymbol{p}^*\|_1\boldsymbol{e}, \boldsymbol{p}^{(1)})}{\sqrt{B_{\Phi}(\mathbf{0}, \boldsymbol{p}^{(1)})}}$, set $E_1 := \{\boldsymbol{e} \in \{0,1\}^m : \langle\mathbf{1}, \boldsymbol{e}\rangle \leq 1\}$.*

The proof idea is to sandwich the cumulative stationarity gap $\sum_{t=1}^{T}\left\langle\boldsymbol{p}^{(t)} - \boldsymbol{p}, \widetilde{\mathbf{g}}^{(t)}\right\rangle$ at a properly chosen anchor point $\boldsymbol{p} := 2\|\boldsymbol{p}^*\|_1\boldsymbol{e}_{j^{\mathcal{A}}} \in \mathcal{P} \cap \mathcal{W}$ (the constant 2 is chosen arbitrarily, it can be any value strictly greater than 1), and then comparing the upper bound and lower bound in the sandwich inequality gives the desired result. The base vector $\boldsymbol{e}_{j^{\mathcal{A}}}$ with a single 1 on $j^{\mathcal{A}}$-th position

indicates which constraint $j^{\mathcal{A}} \in [m]$ is most severely violated. One may notice that the anchor point $\boldsymbol{p} := 2 \|\boldsymbol{p}^*\|_1 \boldsymbol{e}_{j^{\mathcal{A}}}$ needs to be in $\mathcal{P} \cap \mathcal{W}$, which implicitly imposes one additional condition on the domain $\mathcal{W}$ of potential function $\Phi$. While the capability of computationally calculating $\|\boldsymbol{p}^*\|_1$ is not needed for deriving (8), on a practical note, we may need to adjust $\mathcal{W}$ accordingly so that $\mathcal{W}$ contains all possible anchor points $2 \|\boldsymbol{p}^*\|_1 \boldsymbol{e}_{j^{\mathcal{A}}}, \forall j^{\mathcal{A}}$ almost surely. When doing so, we should be very careful because $\boldsymbol{p}^*$ depends on dataset $\mathcal{D}$ and thus, such an adjustment may leak privacy.

Again, one may think $\Phi(\cdot) = \frac{1}{2} \|\cdot\|_2^2$ is the ideal potential function, because its domain $\mathcal{W} = \mathbb{R}^m$ contains all anchor points of interest, does not depend on $\boldsymbol{p}^*$, and therefore no privacy leakage risk. As shown in the first row of Table 2, the guarantees under squared $\ell_2$ are only comparable to, not better than, those in the literature summarized in Table 1. Nevertheless, being comparable is already a significant improvement because our results are derived from weak duality only, whereas existing works all assume strong duality. Moreover, since anchor points $\boldsymbol{p} = 2 \|\boldsymbol{p}^*\|_1 \boldsymbol{e}_{j^{\mathcal{A}}}$ are related to $\ell_1$ norm of $\boldsymbol{p}^*$, if we can find a data-independent upper bound for the value of $\|\boldsymbol{p}^*\|_1$, we might be able to adjust $\mathcal{W}$ accordingly without privacy concerns and may further improve performance. We do so in the next subsection.

Table 2: Examples of theoretical guarantees under specific choices of hyperparameters.

| | Potential Function | | | | Theoretical Guarantees | |
|---|---|---|---|---|---|---|
| | function | domain | strong cvx. $^\dagger$ | init. pt. | opt. gap | total constr. viol. |
| name | $\Phi(\boldsymbol{p})$ | $\mathcal{W}$ | $\alpha$ w.r.t $\|\cdot\|_p$ | $\boldsymbol{p}^{(1)}$ | $\mathsf{F}(\boldsymbol{x}^*) - \mathbb{E}\left[\mathsf{F}(\boldsymbol{x}^{\mathcal{A}})\right]$ | $m\mathbb{E}\left[\left\|\boldsymbol{v}^{\mathcal{A}}\right\|_\infty\right]$ |
| squared $\ell_2$ | $\frac{1}{2}\|\boldsymbol{p}\|_2^2$ | $\mathbb{R}^m$ | 1 w.r.t $\|\cdot\|_2$ | $\frac{1}{\sqrt{m}} \cdot \mathbf{1}$ | $\mathcal{O}\left(\frac{\sqrt{m \ln(1/\delta)}}{\varepsilon}\right)$ | $\mathcal{O}\left(\frac{m^{3/2}\sqrt{\ln(1/\delta)}}{\varepsilon}\right)$ |
| negative entropy (ne) | $\sum_{j=1}^m p_j \ln p_j$ | $\Delta_K$ w. $K = \frac{2\bar{u}}{\gamma \underline{b}}$ | $\frac{1}{K}$ w.r.t. $\|\cdot\|_1$ | $\frac{K}{m} \cdot \mathbf{1}$ | $K\widetilde{\mathcal{O}}\left(\frac{\sqrt{\ln(1/\delta)}}{\varepsilon}\right)$ | $K\widetilde{\mathcal{O}}\left(\frac{m\sqrt{\ln(1/\delta)}}{\varepsilon}\right)$ |
| parameterized ne | $\sum_{j=1}^m b_j p_j \ln(b_j p_j)$ | $\Delta_K(\boldsymbol{b})$ w. $K = \frac{2\bar{u}}{\gamma}$ | $\frac{b_j^2}{K}$ w.r.t. $\|\cdot\|_1$ | $\frac{K}{m b_j}, \forall j$ | $\frac{K}{\underline{b}}\widetilde{\mathcal{O}}\left(\frac{\sqrt{\ln(1/\delta)}}{\varepsilon}\right)$ | $\frac{K}{\underline{b}}\widetilde{\mathcal{O}}\left(\frac{m\sqrt{\ln(1/\delta)}}{\varepsilon}\right)$ |

*Notes.* Results for squared $\ell_2$ (negative entropy) are from Section 3.1 (Section 3.2). Strong duality is necessary for theoretical guarantees by (parameterized) negative entropy. Abbreviations in column titles: cvx.=convexity, init.pt.=initial point, opt.gap = optimality gap, contrs.viol.=constraint violations. The (scaled) simplex set $\Delta_K(\boldsymbol{b})$ with radius $K$ is defined as $\Delta_K(\boldsymbol{b}) := \{\boldsymbol{p} \geq \mathbf{0} : \langle \boldsymbol{b}, \boldsymbol{p} \rangle \leq K\}$; the standard simplex is $\Delta_K := \Delta_K(\mathbf{1})$. $^\dagger$ Squared $\ell_2$ function is strongly convex on whole space $\mathbb{R}^m$, while negative entropy functions only on their respective domains. Results in the **first** and **second** rows are highlighted as the main contributions in Table 1.

## 3.2 Improvements by strong duality

Indeed, there exists space for improvement, if we assume strong duality holds.

**Assumption 3.7** (Strong duality holds). *Strong duality between primal problem* (4) *and dual problem* (5) *holds, i.e.,* $\mathsf{F}(\boldsymbol{x}^*) = \mathsf{D}(\boldsymbol{p}^*)$, *where* $\boldsymbol{x}^*$ *and* $\boldsymbol{p}^*$ *are optimal solutions to* (4) *and* (5), *respectively.*

While weak duality always holds, strong duality does not universally hold. However, there are many applications where strong duality naturally holds. For example, when the dualized constraint $\boldsymbol{a}(\boldsymbol{x}) \leq n\gamma\boldsymbol{b}$ is linear in $\boldsymbol{x}$, strong duality holds. If the constraint is not linear, one can check strong duality by Slater's condition, which essentially says that if the primal problem (4) has strictly feasible solutions, then strong duality holds. The strong duality assumption here is also very mild, and all examples previously considered in the literature satisfy strong duality, for example [HHRW16, Section 4] and [HZ18]. However, these works fail to take full advantage of strong duality. We fill the gap by noticing that strong duality actually restricts $\boldsymbol{p}^*$ to an $\ell_1$ space.

**Lemma 3.8** (Strong duality implies bounded $\boldsymbol{p}^*$). *Let* $\boldsymbol{p}^* := \arg\min_{\boldsymbol{p} \geq \mathbf{0}} \mathsf{D}(\boldsymbol{p})$. *Then, under assumptions 2.5 and 3.7, we have* $\langle \boldsymbol{b}, \boldsymbol{p}^* \rangle \leq \frac{\bar{u}}{\gamma}$ *and* $\langle \mathbf{1}, \boldsymbol{p}^* \rangle \leq \frac{\bar{u}}{\gamma \underline{b}}$ *with* $\underline{b} := \min_j\{b_j\}$.

The above lemma indicates that $\boldsymbol{p}^*$ lies in the interior of a scaled simplex, suggesting the best choice of potential function could be the negative entropy function, which better fits the $\ell_1$ geometry. However, negative entropy $\Phi(\boldsymbol{w}) = \sum_{j=1}^m w_j \ln w_j$ is widely known to be 1-strongly convex only on the probability simplex $\Delta_1 := \{\boldsymbol{w} > \mathbf{0} : \langle \mathbf{1}, \boldsymbol{w} \rangle = 1\}$. To tailor it for our studied problem, we parameterize the negative entropy $\Phi(\boldsymbol{w}; \boldsymbol{\theta}) := \sum_{j=1}^m (\theta_j w_j) \ln(\theta_j w_j)$ with $\boldsymbol{\theta} > \mathbf{0}$, and show that it is also strongly convex *in* a scaled simplex.

**Lemma 3.9** (Strong convexity of parameterized negative entropy). *Let the parameterized negative entropy* $\Phi(\boldsymbol{w}; \boldsymbol{\theta}) := \sum_{j=1}^m (\theta_j w_j) \ln(\theta_j w_j)$ *be defined on* $\mathbb{R}_+^m$, *where we define* $\frac{0}{0} = 0$ *and* $0 \ln 0 = 0$

*by continuity. Then $\Phi(\boldsymbol{w}; \boldsymbol{\theta})$ is $(\min_j\{\theta_j\})^2/K$-strongly convex in $\boldsymbol{w}$ w.r.t. $\|\cdot\|_1$ in a scaled simplex $\Delta_K^{\text{int}}(\boldsymbol{\theta}) := \{\boldsymbol{w} > \boldsymbol{0} : \langle \boldsymbol{\theta}, \boldsymbol{w} \rangle \leq K\}$.*

It is immediate to see that if we set $\boldsymbol{\theta} = \boldsymbol{1}$ and $K = 1$, then Lemma 3.9 recovers the well-known result that negative entropy is 1-strongly convex. For our studied problem, we can let the negative entropy be parameterized by $\boldsymbol{b}$, and use $\Phi(\boldsymbol{w}; \boldsymbol{b}) = \sum_{j=1}^{m}(b_j w_j)\ln(b_j w_j)$ for the Mirror Descent update step (6). This does not compromise privacy, since $\boldsymbol{b}$ is a universal upper bound and is not associated with any specific agent. Now, we are ready to improve the algorithm's performance.

**Theorem 3.10** (Improved utility & feasibility guarantees). *Let $\bar{b} := \max_j\{b_j\}$, $\underline{b} := \min_j\{b_j\}$. Under assumptions 2.5 and 3.7, running algorithm 1 $\mathcal{A}$ with parameterized negative entropy $\Phi(\boldsymbol{w}; \boldsymbol{b})$ with radius $K := 2\bar{u}/\gamma$, $p_j^{(1)} = K/(mb_j), \forall j \in [m]$, and same $T$, $\sigma^2$ as in Theorem 3.5 yields*

$$\text{(utility guarantee)} \quad \mathsf{F}(\boldsymbol{x}^*) - \mathbb{E}_{\mathcal{A}}\left[\mathsf{F}(\boldsymbol{x}^{\mathcal{A}})\right] \leq \frac{8\sqrt{2}\bar{u} \cdot \sqrt{\ln(2m)}}{\gamma\underline{b}} \cdot \sqrt{c_{\varepsilon,\delta}};$$

$$\text{(feasibility guarantee)} \quad \mathbb{E}_{\mathcal{A}}\left[\left\|\boldsymbol{v}^{\mathcal{A}}\right\|_\infty\right] \leq \frac{4\bar{u}\sqrt{2\ln(2m)} \cdot \left[2 + \bar{b} \cdot (\ln(m\bar{b}) - 1)^+\right]}{\gamma\underline{b} \cdot \|\boldsymbol{p}^*\|_1} \cdot \sqrt{c_{\varepsilon,\delta}}.$$

Compared to the results in Theorems 3.5 and 3.6 where both dependencies on number of constraints are $\sqrt{m}$, the results here have better dependencies of $\ln m$. Moreover, following our interpretation of algorithm performance as discussed in the Introduction, Theorem 3.10 implies that the algorithm's ultimate performance is $\widetilde{\mathcal{O}}\left(\sqrt{c_{\varepsilon,\delta}}\right) + m \cdot \widetilde{\mathcal{O}}\left(\sqrt{c_{\varepsilon,\delta}}\right) = \widetilde{\mathcal{O}}\left(m\sqrt{c_{\varepsilon,\delta}}\right)$. The additional $m$ factor comes from $m$ constraints, since the feasibility guarantee in Theorem 3.10 is for any single constraint.

# 4 The lower bound

Some post-processing lemmas for JDP are necessary to prove the lower bound.

**Lemma 4.1** (Self post-processing for JDP). *Let $\mathcal{M} : \mathcal{Z}^n \to \mathcal{X}^n$ be an $(\varepsilon, \delta)$-JDP mechanism and denote its output by $\mathcal{M}(\mathcal{D}) := (\mathcal{M}(\mathcal{D})_1, \ldots, \mathcal{M}(\mathcal{D})_n)$. Let $f : \mathcal{X} \times \mathcal{Z} \to \mathcal{Y}$ be an arbitrary (randomized) function that can be applied element-wisely to $\mathcal{M}$'s output. Then, $(f(\mathcal{M}(\mathcal{D})_1, z_1), \ldots, f(\mathcal{M}(\mathcal{D})_n, z_n))$ is $(\varepsilon, \delta)$-JDP.*

**Lemma 4.2** (Post-processing for JDP). *Let $\mathcal{M} : \mathcal{Z}^n \to \mathcal{X}^n$ be an $(\varepsilon, \delta)$-JDP mechanism and denote its output by $\mathcal{M}(\mathcal{D}) := (\mathcal{M}(\mathcal{D})_1, \ldots, \mathcal{M}(\mathcal{D})_n)$. Let $f : \mathcal{X}^{n-1} \to \mathcal{Y}$ be an arbitrary (randomized) function that can be applied to any collection of $n-1$ elements of $\mathcal{M}$'s output. Then, for any $k \in [n]$, $f(\mathcal{M}(\mathcal{D})_{-k})$ and $f(\mathcal{M}(\mathcal{D}')_{-k})$ are $(\varepsilon, \delta)$-indistinguishable.*[2]

Lemma 4.1 confirms that processing each JDP output with the agent's own data preserves JDP. Lemma 4.2 says applying any operation to $n-1$ elements of two $n$-length JDP outputs obtained from a pair of neighboring datasets will be indistinguishable.

To see how these two lemmas help, we consider the allocations $\boldsymbol{x}^{\mathcal{A}} := (\boldsymbol{x}_1^{\mathcal{A}}, \ldots, \boldsymbol{x}_n^{\mathcal{A}})$ outputted by a JDP algorithm $\mathcal{A} : \mathcal{Z}^n \to \mathcal{X}^n$. By Lemma 4.1, their consumptions $(\boldsymbol{a}_1(\boldsymbol{x}_1^{\mathcal{A}}), \ldots, \boldsymbol{a}_n(\boldsymbol{x}_n^{\mathcal{A}}))$ are also JDP. Furthermore, for any collusive group without agent $k$, their total consumption $\sum_{i \neq k} \boldsymbol{a}_i(\boldsymbol{x}_i^{\mathcal{A}})$ should be insensitive to agent $k$'s allocation by Lemma 4.2. To ensure insensitivity, intuitively but informally, any algorithm $\mathcal{A}$ outputting feasible allocations should *reserve* some resource exclusively for agent $k$. But the reserved resource is wasted in some scenarios, which leads to the lower bound.

**Theorem 4.3** (Minimax lower bound). *For $\varepsilon \geq \max\{1, 1/(n\gamma)\}, 0 < \delta \leq 1/2$, there exists a dataset $\mathcal{D}$ satisfying assumptions 2.5 and 3.7 such that any $(\varepsilon, \delta)$-JDP algorithm $\mathcal{A}$ outputting feasible allocations will lead to a utility loss at least $m/(4\varepsilon)$. Therefore, the minimax lower bound is*

$$\inf_{\mathcal{A} \text{ is } (\varepsilon, \delta)\text{-JDP}} \sup_{\mathcal{D}} \left\{ \mathsf{F}(\boldsymbol{x}^*(\mathcal{D})) - \mathbb{E}_{\mathcal{A}}\left[\mathsf{F}(\boldsymbol{x}^{\mathcal{A}}(\mathcal{D}))\right] \right\} \geq \frac{m}{4\varepsilon}.$$

The lower bound here nearly matches upper bounds given by Theorem 3.10 up to some logarithmic factors in $m$ and $1/\delta$. Therefore, Noisy Dual Mirror Descent is near optimal, under our interpretation

---

[2]Two random variables $X, Y$ are $(\varepsilon, \delta)$-indistinguishable if both $\Pr[X \in \mathcal{S}] \leq e^\varepsilon \Pr[Y \in \mathcal{S}] + \delta$ and $\Pr[Y \in \mathcal{S}] \leq e^\varepsilon \Pr[X \in \mathcal{S}] + \delta$ hold for any subset $\mathcal{S}$.

of algorithm performance. However, the result is limited to $\varepsilon \geq \max\{1, 1/(n\gamma)\}$, and it is interesting to derive lower bounds for other $\varepsilon$. We provide a promising direction in the Appendix, which involves a lower bounding optimization problem with hockey-stick divergence constraints. We conjecture that couplings and divergences employed by [BBG18] are promising tools for this purpose.

## 5 Numerical experiments

### 5.1 Workforce scheduling

Workforce scheduling is about establishing a shift schedule for a given period to maximize workers' total preference while meeting worker availability and shift coverage requirements. Workers' preference and availability are private data to protect. We consider a simple case of scheduling $n = 7$ workers for $m = 14$ days with data publicly available [Opt24]. In this case, (i) each worker has preferences $\boldsymbol{c}_i \in \mathbb{Z}_+^m$, and utility function is $u_i(\boldsymbol{x}_i) = \langle \boldsymbol{c}_i, \boldsymbol{x}_i \rangle$; (ii) shift requirement $r_j \in \mathbb{Z}_+$ is imposed on each day $j \in [m]$, i.e., the number of workers needed for day $j$, which couples all workers; and (iii) workers' availability and shift limits described by a polyhedral set $\mathcal{X}_i := \{\boldsymbol{x}_i \in [0,1]^m : l_i \leq \langle \boldsymbol{1}, \boldsymbol{x}_i \rangle \leq u_i\}$. The problem is modeled as $\max\{\sum_{i=1}^n \langle \boldsymbol{c}_i, \boldsymbol{x}_i \rangle : \sum_i \boldsymbol{x}_i = \boldsymbol{r}; \boldsymbol{x}_i \in \mathcal{X}_i, \forall i\}$. While this is a small case, it is suitable for visualizing decisions and understanding the impact of JDP. Moreover, since the optimal non-private $\boldsymbol{p}^* = [0, 3, 1, 0, 2, 0, 0, 4, 3, 2, 3, 0, 0, 0]$ is a sparse vector, one can expect MD equipped with negative entropy to perform better.

Figure 1 shows the final decisions under various $\varepsilon$, where the rightmost is the optimal non-private decision where a value of 1 means arranging the person to that day. Private decisions are fractional

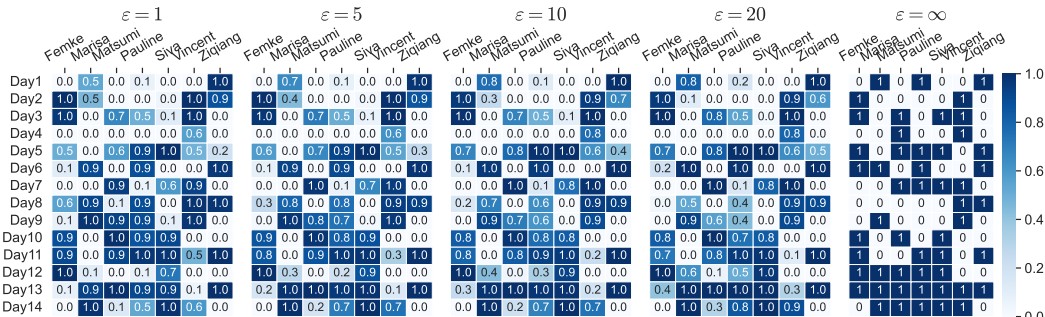

Figure 1: 7-person, 14-day rosters under various $\varepsilon$. Algorithms parameters: potential function is negative entropy parameterized by $\boldsymbol{b} = \boldsymbol{1}$; $K = 1.1\bar{u}/(\gamma\underline{b})$, $\delta = .01$, $T = 10^4$. Other settings follow Theorem 3.10. Results reported are averages of 50 runs. Strong duality holds due to linearity.

and can be interpreted as probabilities. It is not hard to see that private decisions are in a similar pattern to non-private decisions, and many private decisions are exactly the same as their optimal non-private counterpart. Moreover, private decisions different from their non-private counterpart converge gradually as $\varepsilon \to \infty$, see for example, the decision of (Day1, Marisa), (Day2, Marisa), and (Day4, Vincent). Optimality gaps and constraint violations are shown in Figure 2 (left panel). For all

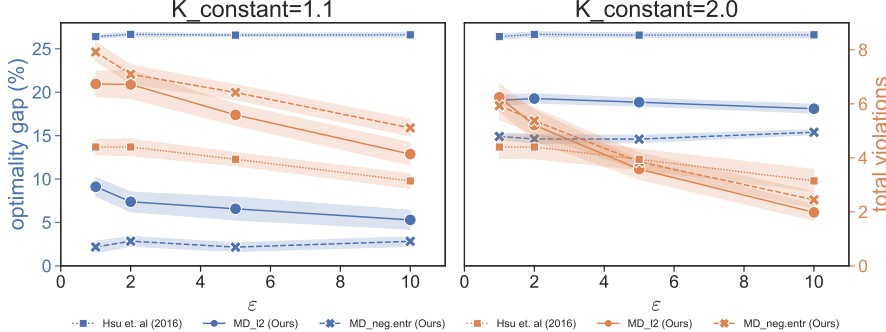

Figure 2: $\varepsilon$ v.s.optimality gaps & constraint violations. More discussions in Appendix C.

curves, the lower the better. It is clear that our algorithms have significantly smaller optimality gaps at slightly higher constraint violations.

One may notice that the constant of $K$ is 1.1 rather than 2 suggested by Theorem 3.10. This is the consequence of parameter tuning, and we show its impact in Figure 2 (right panel). Comparing the two plots in Figure 2, it is obvious that larger K_constant leads to conservative decisions: fewer constraint violations but higher optimality gaps. Moreover, when K_constant is 2, our algorithms achieve lower optimality gaps at the same level of constraint violations.

## 5.2  Assignment problem

We next consider three large-scale assignment problems with $(m, n)$ = (8, 800), (15, 1500), (30, 3000), aiming to maximize aggregated utility by assigning at most one unit to each agent. The problem is modeled as $\max \{\sum_{i=1}^{n} \langle c_i, x_i \rangle : \sum_i I_{m \times n} x_i \leq n\gamma \mathbf{1}; x_i \in \mathcal{X}_i, \forall i\}$ with $\mathcal{X}_i := \{x_i \in [0,1]^n : \langle \mathbf{1}, x_i \rangle \leq 1\}$ being a unit simplex and $I_{m \times n} := [I_m, \ldots, I_m]$ being a consumption matrix concatenated from $n/m$ identity matrices. Raw data of $c_i$ are available at [Bea04]. In this assignment problem, $p^* > 0$ is strictly non-zero. We run experiments and report results in Table 3.

Table 3: Algorithm performance, mean±sd

|  |  | Optimality gap $\frac{F(x^*)-F(x^{\mathcal{A}})}{F(x^*)} \times 100\%$ | | | | Total constraint violation $\|v^{\mathcal{A}}\|_1$ | | | |
|---|---|---|---|---|---|---|---|---|---|
| $n$ | Alg. | $\varepsilon = 1$ | 2 | 5 | 10 | $\varepsilon = 1$ | 2 | 5 | 10 |
| 800 | [HHRW16] | **-1.8**±2.6 | 4.3±3.2 | 5.3±2.8 | 3.8±1.5 | 84.8±18.4 | 46.2±16.8 | 19.0±9.3 | 11.0±5.2 |
|  | MD_l2 (ours) | 1.8±3.9 | **1.3**±2.3 | 0.8±0.8 | 0.5±0.5 | 27.7±12.0 | 12.7±8.3 | 5.1±2.6 | 2.7±1.6 |
|  | MD_ne (ours) | 2.1±4.0 | 2.0±2.2 | **0.7**±0.9 | **0.4**±0.5 | **23.9**±13.3 | **9.2**±6.7 | **4.7**±3.4 | **2.6**±1.5 |
| 1500 | [HHRW16] | **-6.9**±1.5 | 1.8±1.4 | 13.8±1.3 | 12.3±0.9 | 256.7±16.0 | 156.8±20.8 | 30.3±10.8 | 15.7±6.7 |
|  | MD_l2 (ours) | 4.3±3.4 | 2.8±2.3 | 1.2±0.8 | 0.7±0.7 | 54.6±18.5 | 35.1±14.8 | **12.6**±4.6 | 7.2±3.3 |
|  | MD_ne (ours) | 6.0±4.1 | 3.0±2.1 | **1.1**±0.9 | **0.6**±0.6 | **47.8**±23.0 | **25.8**±12.0 | 13.4±5.9 | **4.5**±2.5 |
| 3000 | [HHRW16] | **-29.4**±0.9 | **-18.9**±1.2 | 6.9 ±1.2 | 8.2±1.0 | $> 10^3$ | $> 10^3$ | 743.8±25.7 | 680.1±29.4 |
|  | MD_l2 (ours) | 4.3±5.1 | 3.0±2.4 | **2.0**±1.3 | 1.5±0.7 | 193.9±54.1 | 97.2±23.9 | 39.8±12.6 | 22.0±6.3 |
|  | MD_ne (ours) | 10.4±5.2 | 6.1±3.0 | 3.0±1.1 | **1.6**±0.6 | **134.5**±56.4 | **65.4**±25.1 | **27.9**±8.4 | **18.1**±6.0 |

*Notes.* Parameters for our algorithms: $K = 1.1\bar{u}/(\gamma\underline{b})$, $\delta = .01$, $T = 10^4$. Other settings follow Theorem 3.10. MD_l2 and MD_ne mean that potential function is squared $\ell_2$ and negative entropy, respectively. For three cases $n = 800, 1500, 3000$, we set resource level $\gamma = 0.1, 0.05, 0.02$, respectively. For all values in the table, the lower the better. **Bold**=better. More results are deferred to Appendix C.

From the table, we observe MD_l2 has lower gaps and reasonable constraint violations for small $\varepsilon$; and MD_ne almost dominates others for large $\varepsilon$. Both our algorithms outperform existing methods.

# 6  Discussion and conclusion

**Limitation of our work.**    One significant limitation is the restriction of $\varepsilon \geq \max\{1, 1/(n\gamma)\}$ to make the lower bound hold. However, we identify a promising direction to overcome this limitation in Appendix B.4. Once the limitation is overcame, the analysis may further uncover lower bounds on online cases [SS18], a long-standing open question in private online optimization [CDE$^+$24].

In this work, we considered convex resource allocation problems under joint differential privacy. To solve the problem approximately, we proposed an algorithm Noisy Dual Mirror Descent, which privatizes dual variables of hard constraints, and then uses private dual variables to coordinate allocations. A significant merit of the algorithm is its ability to better leverage the geometric structure of the dual space; thus, it is provably near optimal for a large range of privacy parameters. There are many interesting directions for future study. For example, getting rid of the limitation discussed previously is a fruitful endeavour. Identifying a proper DP manner to tune the constant of $K$ is of more practical interest.

## Acknowledgments and Disclosure of Funding

We sincerely thank the anonymous Area Chair and reviewers for their valuable feedback that significantly improved this work. This research is supported by the Ministry of Education, Singapore, under its MOE AcRF Tier 1 (RG117/23).

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

# A Proofs for Section 3

## A.1 Proof of Theorem 3.2

*Proof.* Since each individual's allocation $x_i^A$ depends on all shadow prices $(p^{(1)}, \dots, p^{(T)})$ on the training trajectory and the individual's own data, by Billboard Lemma 2.3, it suffices to verify the whole sequence $(p^{(1)}, \dots, p^{(T)})$ is $(\varepsilon, \delta)$-DP. We characterize the DP guarantee through Rényi Differential Privacy [Mir17]. Below are some useful lemmas on Rényi DP.

**Definition A.1** $((\alpha, \varepsilon)$-Rényi Differential Privacy, [Mir17]). *Let $\mathcal{M} : \mathcal{D} \to \mathcal{P}$ be a randomized mechanism. For any neighboring datasets $\mathcal{D} \sim \mathcal{D}'$, let $\mathbb{P}, \mathbb{P}'$ be the distribution of $\mathcal{M}(\mathcal{D})$ and $\mathcal{M}(\mathcal{D}')$, respectively. Then, $\mathcal{M}$ is said to be $\varepsilon$-Rényi differentially private of order $\alpha$, or $(\alpha, \varepsilon)$-RDP for short, if it holds that*

$$D_\alpha(\mathbb{P} \,||\, \mathbb{P}') \le \varepsilon, \quad \forall \mathcal{D} \sim \mathcal{D}',$$

*where $D_\alpha(\mathbb{P} \,||\, \mathbb{Q}) := \frac{1}{\alpha-1} \ln \left( \mathbb{E}_{\mathbb{Q}} \left[ \left( \frac{\mathbb{P}(x)}{\mathbb{Q}(x)} \right)^\alpha \right] \right)$ is the Rényi divergence between $\mathbb{P}$ and $\mathbb{Q}$.*

**Lemma A.2** (Gaussian Mechanism is RDP, Proposition 7 in [Mir17]). *Let $f : \mathcal{D} \to \mathcal{Z}$ be a vector-valued function whose global sensitivity is $GS := \sup_{\mathcal{D} \sim \mathcal{D}'} \|f(\mathcal{D}) - f(\mathcal{D}')\|_2$, then the Gaussian Mechanism $\mathcal{M}(\mathcal{D}) := f(\mathcal{D}) + \mathcal{N}(\mathbf{0}, \sigma^2 \mathbf{I})$ is $(\alpha, \frac{\alpha \cdot GS^2}{2\sigma^2})$-RDP, for any $\alpha > 1$.*

**Lemma A.3** (Composition for RDP, Proposition 1 in [Mir17]). *Let $\mathcal{M}_1 : \mathcal{D} \to \mathcal{P}_1$ be $(\alpha, \varepsilon_1)$-RDP and $\mathcal{M}_2 : \mathcal{D} \times \mathcal{P}_1 \to \mathcal{P}_2$ be $(\alpha, \varepsilon_2)$-RDP. Let $X := \mathcal{M}_1(\mathcal{D})$ be the output of $\mathcal{M}_1$ and $Y := \mathcal{M}_2(\mathcal{D}, X)$ be the output of $\mathcal{M}_2$, then the adaptive composition $(X, Y)$ satisfies $(\alpha, \varepsilon_1 + \varepsilon_2)$-RDP.*

**Lemma A.4** (From RDP to DP, Proposition 3 in [Mir17]). *If $\mathcal{M}$ is an $(\alpha, \varepsilon)$-RDP mechanism, it also satisfies $(\varepsilon + \frac{\ln(1/\delta)}{\alpha-1}, \delta)$-DP, for any $\delta \in (0, 1)$.*

For iteration $t$, we notice that data only plays a role in the gradient $\mathbf{g}(p^{(t)}; \mathcal{D}) := n\gamma b - a(x(p^{(t)}; \mathcal{D}))$ through intermediate allocation decision $x(p^{(t)}; \mathcal{D}) := \arg\max_{x \in \mathcal{X}} \mathsf{F}(x) - \langle p^{(t)}, a(x) \rangle$. It is immediate to show that the Global Sensitivity of gradient function is

$$\mathrm{GS} = \sup_{p, \mathcal{D} \sim \mathcal{D}'} \|\mathbf{g}(p; \mathcal{D}) - \mathbf{g}(p; \mathcal{D}')\|_2 = \sup_{p; z_i \sim z_i'} \|a_i(x_i(p; z_i)) - a_i'(x_i(p; z_i'))\|_2 \le \|b\|_2, \quad (9)$$

where the last inequality is by our assumption that serving one individual consumes at most $b$ resource (see assumption 2.5). Then, injecting a zero-mean Gaussian noise $\mathbf{n}^{(t)} \sim \mathcal{N}(\mathbf{0}, \sigma^2 \mathbf{I})$ into gradients $\mathbf{g}^{(t)} := \mathbf{g}(p^{(t)}; \mathcal{D})$ can guarantee $(\alpha, \frac{\alpha \cdot \|b\|_2^2}{2\sigma^2})$-RDP by Lemma A.2.

Moreover, the algorithm comprises $T$ iterations, and each iteration takes as input the shadow price in previous iteration; therefore, $T$-fold iterations form a sequential composition. Applying Composition Lemma A.3, we immediately conclude that $(\mathbf{g}^{(1)} + \mathbf{n}^{(1)}, \dots, \mathbf{g}^{(T)} + \mathbf{n}^{(T)})$ satisfies $(\alpha, \frac{T\alpha \cdot \|b\|_2^2}{2\sigma^2})$-RDP. As the noisy mirror descent update step in Eq.(6) is merely post-processing after obtaining $(\mathbf{g}^{(1)} + \mathbf{n}^{(1)}, \dots, \mathbf{g}^{(T)} + \mathbf{n}^{(T)})$, the sequence $(p^{(1)}, \dots, p^{(T)})$ is thus also $(\alpha, \frac{T\alpha \cdot \|b\|_2^2}{2\sigma^2})$-RDP. Lastly, by Lemma A.4, $(\alpha, \frac{T\alpha \cdot \|b\|_2^2}{2\sigma^2})$-RDP implies $(\frac{T\alpha \cdot \|b\|_2^2}{2\sigma^2} + \frac{\ln(1/\delta)}{\alpha-1}, \delta)$-DP $\forall \delta \in (0, 1)$. Thus, for a given pair of $(\varepsilon, \delta)$, if we assign privacy budget in the following way:

$$\begin{cases} \frac{T\alpha\|b\|_2^2}{2\sigma^2} = \varepsilon/2; \\ \frac{\ln(1/\delta)}{\alpha-1} = \varepsilon/2, \end{cases}$$

we can get

$$\sigma^2 = T \|b\|_2^2 \cdot \left( \frac{2\ln(1/\delta)}{\varepsilon^2} + \frac{1}{\varepsilon} \right),$$

which is the desired variance level.

$\square$

## A.2 Proof of Lemma 3.4

*Proof.* For any gradient $\widetilde{\mathbf{g}}^{(t)}$ used for the mirror descent update step Eq.(6), shadow prices $\boldsymbol{p}^{(t)}$, and updated shadow prices $\boldsymbol{p}^{(t+1)}$, there exists an inequality linking them together (Lemma 9.13, [Bec17]):

$$\left\langle \widetilde{\mathbf{g}}^{(t)}, \boldsymbol{p}^{(t)} - \boldsymbol{p} \right\rangle \leq \frac{\eta^{(t)} \left\| \widetilde{\mathbf{g}}^{(t)} \right\|_q^2}{2\alpha} + \frac{1}{\eta^{(t)}} \left\{ B_\Phi(\boldsymbol{p}, \boldsymbol{p}^{(t)}) - B_\Phi(\boldsymbol{p}, \boldsymbol{p}^{(t+1)}) \right\}, \quad \forall t, \quad \forall \boldsymbol{p} \in \mathcal{P} \cap \mathcal{W}.$$

Summing up the above inequality in a telescoping way over $t = 1, \ldots, T$ gives

$$\sum_{t=1}^T \left\langle \widetilde{\mathbf{g}}^{(t)}, \boldsymbol{p}^{(t)} - \boldsymbol{p} \right\rangle \leq \frac{\sum_{t=1}^T \eta^{(t)} \left\| \widetilde{\mathbf{g}}^{(t)} \right\|_q^2}{2\alpha} + \frac{B_\Phi(\boldsymbol{p}, \boldsymbol{p}^{(1)})}{\eta^{(1)}} - \frac{B_\Phi(\boldsymbol{p}, \boldsymbol{p}^{(T+1)})}{\eta^{(T)}}$$
$$+ \sum_{t=1}^{T-1} \left( \frac{1}{\eta^{(t+1)}} - \frac{1}{\eta^{(t)}} \right) B_\Phi(\boldsymbol{p}, \boldsymbol{p}^{(t+1)}).$$

When stepsizes $\eta^{(t)} = \eta$ are the same for all $t$, the inequality becomes

$$\sum_{t=1}^T \left\langle \widetilde{\mathbf{g}}^{(t)}, \boldsymbol{p}^{(t)} - \boldsymbol{p} \right\rangle \leq \frac{\eta \sum_{t=1}^T \left\| \widetilde{\mathbf{g}}^{(t)} \right\|_q^2}{2\alpha} + \frac{B_\Phi(\boldsymbol{p}, \boldsymbol{p}^{(1)})}{\eta}.$$

$\square$

## A.3 Proof of Theorem 3.5

*Proof.* Let us first fix a sequence of noises $\{\mathbf{n}^{(t)}\}_{t=1}^T$. Recall that the final allocation decision $\boldsymbol{x}^{\mathcal{A}}$ is the average over all decisions along the trajectory and the objective function is concave; therefore, by Jensen's inequality, $\mathsf{F}(\boldsymbol{x}^{\mathcal{A}}) \geq \frac{1}{T} \sum_{t=1}^T \mathsf{F}(\boldsymbol{x}^{(t)})$. Moreover, by weak duality, $\mathsf{F}(\boldsymbol{x}^*) \leq \mathsf{D}(\boldsymbol{p}^{(t)}), \forall t$. With the above results, the optimality gap $\mathsf{F}(\boldsymbol{x}^*) - \mathsf{F}(\boldsymbol{x}^{\mathcal{A}})$ can be upper controlled as:

$$\mathsf{F}(\boldsymbol{x}^*) - \mathsf{F}(\boldsymbol{x}^{\mathcal{A}}) \leq \frac{1}{T} \sum_{t=1}^T \left[ \mathsf{D}(\boldsymbol{p}^{(t)}) - \mathsf{F}(\boldsymbol{x}^{(t)}) \right]$$
$$= \frac{1}{T} \sum_{t=1}^T \left[ \mathsf{F}(\boldsymbol{x}^{(t)}) + \left\langle \boldsymbol{p}^{(t)}, \mathbf{g}^{(t)} \right\rangle - \mathsf{F}(\boldsymbol{x}^{(t)}) \right] \qquad \text{(by definition of dual problem D)}$$
$$= \frac{1}{T} \sum_{t=1}^T \left\langle \widetilde{\mathbf{g}}^{(t)}, \boldsymbol{p}^{(t)} - \mathbf{0} \right\rangle - \frac{1}{T} \sum_{t=1}^T \left\langle \mathbf{n}^{(t)}, \boldsymbol{p}^{(t)} \right\rangle.$$

Taking expectation with respect to noises $\{\mathbf{n}^{(t)}\}_{t=1}^T$ on both sides, we can remove the second term on the right hand side in preceding inequality, because $\mathbf{n}^{(t)}$ is zero-mean and is independent of $\boldsymbol{p}^{(t)}$. This leads to

$$\mathsf{F}(\boldsymbol{x}^*) - \mathbb{E}_{\mathcal{A}}\left[ \mathsf{F}(\boldsymbol{x}^{\mathcal{A}}) \right] \leq \frac{1}{T} \cdot \mathbb{E}_{\mathcal{A}}\left[ \sum_{t=1}^T \left\langle \widetilde{\mathbf{g}}^{(t)}, \boldsymbol{p}^{(t)} - \mathbf{0} \right\rangle \right].$$

The inner summation can be upper bounded by applying Lemma 3.4 with $\boldsymbol{p} = \mathbf{0}$ and $\eta^{(t)} = \eta > 0, \forall t$; we therefore obtain

$$\mathsf{F}(\boldsymbol{x}^*) - \mathbb{E}_{\mathcal{A}}\left[ \mathsf{F}(\boldsymbol{x}^{\mathcal{A}}) \right] \leq \mathbb{E}_{\mathcal{A}}\left[ \frac{\eta \sum_{t=1}^T \left\| \widetilde{\mathbf{g}}^{(t)} \right\|_q^2}{2\alpha T} + \frac{1}{\eta T} B_\Phi(\mathbf{0}, \boldsymbol{p}^{(1)}) \right]. \qquad (10)$$

Here are some helpful inequalities:

$$\left\| \widetilde{\mathbf{g}}^{(t)} \right\|_q^2 = \left\| \mathbf{g}^{(t)} + \mathbf{n}^{(t)} \right\|_q^2 \leq 2 \left\| \mathbf{g}^{(t)} \right\|_q^2 + 2 \left\| \mathbf{n}^{(t)} \right\|_q^2;$$
$$\left\| \mathbf{g}^{(t)} \right\|_q = \left\| \gamma n \boldsymbol{b} - \boldsymbol{a}(\boldsymbol{x}^{(t)}) \right\|_q \leq \bar{\gamma} n \left\| \boldsymbol{b} \right\|_q.$$

Denote $G := \bar{\gamma}^2 n^2 \|\boldsymbol{b}\|_q^2$, and $G_{\mathcal{A}} := \frac{1}{T} \sum_{t=1}^{T} \|\mathbf{n}^{(t)}\|_q^2$. Using above helpful inequalities, we can upper bound $\frac{1}{T} \sum_{t=1}^{T} \|\widetilde{\mathbf{g}}^{(t)}\|_q^2$ as

$$\frac{1}{T} \sum_{t=1}^{T} \left\| \widetilde{\mathbf{g}}^{(t)} \right\|_q^2 \leq 2G + 2G_{\mathcal{A}}. \tag{11}$$

Plugging (11) back into (10), we further reach an upper bound:

$$\mathsf{F}(\boldsymbol{x}^*) - \mathbb{E}_{\mathcal{A}} \left[ \mathsf{F}(\boldsymbol{x}^{\mathcal{A}}) \right] \leq \frac{\eta \cdot (G + \mathbb{E}_{\mathcal{A}}[G_{\mathcal{A}}])}{\alpha} + \frac{1}{\eta T} B_{\Phi}(\boldsymbol{0}, \boldsymbol{p}^{(1)}).$$

Because $G_{\mathcal{A}} = \frac{1}{T} \sum_{t=1}^{T} \|\mathbf{n}^{(t)}\|_q^2$ and $\mathbf{n}^{(t)} \sim \mathcal{N}(\boldsymbol{0}, \sigma^2 \boldsymbol{I}_{m \times m})$ with $\sigma^2 = T c_{\varepsilon, \delta}, \forall t$, direct calculation gives $\mathbb{E}_{\mathcal{A}}[G_{\mathcal{A}}] = \sigma^2 \mathbb{E}\left[\|\boldsymbol{z}\|_q^2\right]$ with $\boldsymbol{z} \sim \mathcal{N}(\boldsymbol{0}, \boldsymbol{I}_{m \times m})$ being a standard Gaussian random vector. Therefore,

$$\mathsf{F}(\boldsymbol{x}^*) - \mathbb{E}_{\mathcal{A}} \left[ \mathsf{F}(\boldsymbol{x}^{\mathcal{A}}) \right] \leq \frac{\eta \cdot \left( G + \sigma^2 \mathbb{E}\left[\|\boldsymbol{z}\|_q^2\right] \right)}{\alpha} + \frac{1}{\eta T} B_{\Phi}(\boldsymbol{0}, \boldsymbol{p}^{(1)}).$$

Let $C_{\Phi}(\boldsymbol{p}^{(1)}) := \sqrt{B_{\Phi}(\boldsymbol{0}, \boldsymbol{p}^{(1)})/\alpha}$. Then, plugging in the stepsize $\eta = \sqrt{\frac{\alpha B_{\Phi}(\boldsymbol{0}, \boldsymbol{p}^{(1)})}{T \cdot (G + \sigma^2 \mathbb{E}[\|\boldsymbol{z}\|_q^2])}}$ leads to

$$\mathsf{F}(\boldsymbol{x}^*) - \mathbb{E}_{\mathcal{A}} \left[ \mathsf{F}(\boldsymbol{x}^{\mathcal{A}}) \right] \leq 2 \sqrt{\frac{\left( G + \sigma^2 \mathbb{E}\left[\|\boldsymbol{z}\|_q^2\right] \right) \cdot B_{\Phi}(\boldsymbol{0}, \boldsymbol{p}^{(1)})}{\alpha T}}$$

$$\leq 2 C_{\Phi}(\boldsymbol{p}^{(1)}) \cdot \left( \frac{\sqrt{G}}{\sqrt{T}} + \sqrt{c_{\varepsilon, \delta}} \cdot \sqrt{\mathbb{E}\left[\|\boldsymbol{z}\|_q^2\right]} \right). \quad \text{(by } \sigma^2 = T c_{\varepsilon, \delta} \text{)} \tag{12}$$

Lastly, if we run sufficiently many iterations $T \geq \frac{G}{c_{\varepsilon, \delta} \cdot \mathbb{E}[\|\boldsymbol{z}\|_q^2]}$, we get the stated result in the theorem.

$\square$

## A.4 Proof of Theorem 3.6

*Proof.* We first briefly introduce the idea for the proof. We will find a lower bound and an upper bound for the cumulative stationarity gap $\sum_{t=1}^{T} \langle \widetilde{\mathbf{g}}^{(t)}, \boldsymbol{p}^{(t)} - \boldsymbol{p} \rangle$ in Lemma 3.4. With a properly chosen $\boldsymbol{p} \in \mathbb{R}_+^m \cap \mathcal{W}$ as a bridge, comparing these two bounds leads to the desired conclusion.

For a given final allocation decision $\boldsymbol{x}^{\mathcal{A}}$, we denote the resource underage level by

$$\boldsymbol{v}^{\mathcal{A}} := \left( \boldsymbol{a}(\boldsymbol{x}^{\mathcal{A}}) - n\gamma \boldsymbol{b} \right)^+,$$

where the positive part operator $(\cdot)^+$ applies element-wise. It is self-evident that $\boldsymbol{v}^{\mathcal{A}} \geq 0$ by definition, and $\boldsymbol{v}^{\mathcal{A}} = \boldsymbol{0}$ implies no constraint violation. Let $j^{\mathcal{A}} := \arg\max_{j \in [m] \cup \{0\}} \{u_j^{\mathcal{A}}\}$ be the index of the most-severely violated constraint. Specially, if $j^{\mathcal{A}} = 0$, then we know $\boldsymbol{v}^{\mathcal{A}} = \boldsymbol{0}$, and we are done. Otherwise, we let $\boldsymbol{e}_{j^{\mathcal{A}}}$ be an indicator vector with 1 on its $(j^{\mathcal{A}})$'s position and 0 on others.

Choose $\boldsymbol{p} := C \boldsymbol{e}_{j^{\mathcal{A}}} \in \mathbb{R}_+^m \cap \mathcal{W}$ with a constant $C$ to be determined later, we reformulate the cumulative stationarity gap $\sum_{t=1}^{T} \langle \widetilde{\mathbf{g}}^{(t)}, \boldsymbol{p}^{(t)} - C \boldsymbol{e}_{j^{\mathcal{A}}} \rangle$ into

$$\sum_{t=1}^{T} \left\langle \widetilde{\mathbf{g}}^{(t)}, \boldsymbol{p}^{(t)} - C \boldsymbol{e}_{j^{\mathcal{A}}} \right\rangle = \underbrace{\sum_{t=1}^{T} \left\langle \mathbf{g}^{(t)}, \boldsymbol{p}^{(t)} \right\rangle}_{(*)} + \underbrace{\sum_{t=1}^{T} \left\langle \mathbf{g}^{(t)}, -C \boldsymbol{e}_{j^{\mathcal{A}}} \right\rangle}_{(**)} + \underbrace{\sum_{t=1}^{T} \left\langle \mathbf{n}^{(t)}, \boldsymbol{p}^{(t)} - C \boldsymbol{e}_{j^{\mathcal{A}}} \right\rangle}_{(***)},$$

and control the three terms one by one.

**(\*):** First, rewrite $(*)$ into a function of dual form:

$$(*) = \sum_{t=1}^{T} \left\{ \left[ \mathsf{F}(\boldsymbol{x}^{(t)}) + \left\langle \boldsymbol{p}^{(t)}, \mathbf{g}^{(t)} \right\rangle \right] - \mathsf{F}(\boldsymbol{x}^{(t)}) \right\} = \sum_{t=1}^{T} \left[ \mathsf{D}(\boldsymbol{p}^{(t)}) - \mathsf{F}(\boldsymbol{x}^{(t)}) \right] \geq T \cdot \left[ \mathsf{D}\left(\boldsymbol{p}^{*}\right) - \mathsf{F}(\boldsymbol{x}^{\mathcal{A}}) \right],$$

(13)

where the last inequality is from $\mathsf{D}(\boldsymbol{p}^{(t)}) \geq \mathsf{D}(\boldsymbol{p}^{*})$ and applying Jensen's inequality to $\mathsf{F}(\cdot)$. Next, we turn to control $\mathsf{F}(\boldsymbol{x}^{\mathcal{A}})$. We observe that if more resource $\boldsymbol{v}^{\mathcal{A}}$ becomes available for the original allocation problem, then $\boldsymbol{x}^{\mathcal{A}}$ will be a feasible allocation. Hence, we can upper bound the value of $\mathsf{F}(\boldsymbol{x}^{\mathcal{A}})$ with another problem having more resources. Denote the set of feasible allocations when $\boldsymbol{b}$ resource available by $\mathcal{X}(\boldsymbol{b}) := \{\boldsymbol{x} := (\boldsymbol{x}_1, \ldots, \boldsymbol{x}_n) \in \otimes_{i=1}^{n} \mathcal{X}_i : \boldsymbol{a}(\boldsymbol{x}) \leq \boldsymbol{b}\}$. We have

$$\mathsf{F}(\boldsymbol{x}^{\mathcal{A}}) \leq \max_{\boldsymbol{x} \in \mathcal{X}(n\gamma\boldsymbol{b} + \boldsymbol{v}^{\mathcal{A}})} \mathsf{F}(\boldsymbol{x}) \qquad\qquad (\boldsymbol{x}^{\mathcal{A}} \text{ is feasible})$$

$$\leq \max_{\boldsymbol{x} \in \otimes_{i=1}^{n} \mathcal{X}_i} \left\{ \mathsf{F}(\boldsymbol{x}) + \left\langle \boldsymbol{p}, n\gamma\boldsymbol{b} + \boldsymbol{v}^{\mathcal{A}} - \boldsymbol{a}(\boldsymbol{x}) \right\rangle \right\} \qquad (\text{weak duality}, \forall \boldsymbol{p} \in \mathcal{P})$$

$$= \mathsf{D}(\boldsymbol{p}) + \left\langle \boldsymbol{p}, \boldsymbol{v}^{\mathcal{A}} \right\rangle. \qquad\qquad\qquad (\text{dual form}) \qquad (14)$$

Let $\boldsymbol{p}$ in (14) be $\boldsymbol{p} := \boldsymbol{p}^{*}$, and plug (14) back into (13), we obtain

$$(*) \geq -T \cdot \left\langle \boldsymbol{p}^{*}, \boldsymbol{v}^{\mathcal{A}} \right\rangle \geq -T \left\| \boldsymbol{v}^{\mathcal{A}} \right\|_{\infty} \cdot \left\| \boldsymbol{p}^{*} \right\|_1.$$

**(\*\*):** The second term $(**)$ helps characterize constraint violations:

$$(**) = C \sum_{t=1}^{T} \left\langle -\mathbf{g}^{(t)}, \boldsymbol{e}_{j^{\mathcal{A}}} \right\rangle = CT \cdot \left\langle \frac{1}{T} \sum_{t=1}^{T} \boldsymbol{a}(\boldsymbol{x}^{(t)}) - n\gamma\boldsymbol{b}, \boldsymbol{e}_{j^{\mathcal{A}}} \right\rangle$$

$$\geq CT \cdot \left\langle \boldsymbol{a}(\boldsymbol{x}^{\mathcal{A}}) - n\gamma\boldsymbol{b}, \boldsymbol{e}_{j^{\mathcal{A}}} \right\rangle \qquad (\text{by assumption } \boldsymbol{a}(\cdot) \text{ is convex})$$

$$= CT \left\| \boldsymbol{v}^{\mathcal{A}} \right\|_{\infty}. \qquad\qquad (\text{by definitions of } \boldsymbol{v}^{\mathcal{A}} \text{ and } \boldsymbol{e}_{j^{\mathcal{A}}})$$

**(\*\*\*):** The third term $(***)$ involves zero-mean Gaussian noise vectors $\{\mathbf{n}^{(t)}\}_t$ that are independent of $\boldsymbol{p}^{(t)}$. While $\boldsymbol{e}_{j^{\mathcal{A}}}$ is dependent on noise vectors, the dependence is not a big issue here, since $\boldsymbol{e}_{j^{\mathcal{A}}} \in E_1 := \{\boldsymbol{e} \in \{0,1\}^m : \langle \boldsymbol{e}, \mathbf{1} \rangle = 1\}$, and the Gaussian Complexity $\mathcal{G}(E_1) := \mathbb{E}_{\boldsymbol{z} \sim \mathcal{N}(\mathbf{0}, \boldsymbol{I}_{m \times m})} \left[ \sup_{\boldsymbol{e} \in E_1} \langle \boldsymbol{e}, \boldsymbol{z} \rangle \right]$ of the set $E_1$ can be well controlled:

$$\mathcal{G}(E_1) = \mathbb{E}_{\boldsymbol{z} \sim \mathcal{N}(\mathbf{0}, \boldsymbol{I}_{m \times m})} \left[ \max\{z_1, \ldots, z_m\} \right] \leq \sqrt{2 \ln m}.$$

Therefore, the term $(***)$ admits a lower bound in expectation w.r.t. $\mathcal{A}$ as shown below:

$$\mathbb{E}_{\mathcal{A}} \left[ (***) \right] = 0 - C \cdot \mathbb{E}_{\mathcal{A}} \left[ \left\langle \sum_{t=1}^{T} \mathbf{n}^{(t)}, \boldsymbol{e}_{j^{\mathcal{A}}} \right\rangle \right] \geq -C \cdot \mathbb{E}_{\mathcal{A}} \left[ \sup_{\boldsymbol{e} \in E_1} \left\langle \sum_{t=1}^{T} \mathbf{n}^{(t)}, \boldsymbol{e} \right\rangle \right]$$

$$= -C\sqrt{T}\sigma \cdot \mathcal{G}(E_1)$$

$$\geq -C\sqrt{T}\sigma \cdot \sqrt{2 \ln m}$$

$$= -CT \cdot \sqrt{c_{\varepsilon,\delta} \cdot 2 \ln m}. \qquad (\text{by } \sigma = \sqrt{T c_{\varepsilon,\delta}})$$

Finally, replacing three terms back and taking expectation w.r.t. $\mathcal{A}$, we obtain a lower bound on the cumulative stationarity gap:

$$\mathbb{E}_{\mathcal{A}} \left[ \sum_{t=1}^{T} \left\langle \widetilde{\mathbf{g}}^{(t)}, \boldsymbol{p}^{(t)} - C\boldsymbol{e}_{j^{\mathcal{A}}} \right\rangle \right] \geq T\mathbb{E}_{\mathcal{A}} \left[ \left\| \boldsymbol{v}^{\mathcal{A}} \right\|_{\infty} \right] \cdot (C - \left\| \boldsymbol{p}^{*} \right\|_1) - CT \cdot \sqrt{c_{\varepsilon,\delta} \cdot 2 \ln m}. \quad (15)$$

Now, we turn to another side and bound the cumulative stationarity gap from above. Suppose we can choose a proper $C > 0$ such that $C\boldsymbol{e}_{j^{\mathcal{A}}} \in \mathbb{R}_+^m \cap \mathcal{W}$ almost surely, the expected cumulative stationarity gap thus can be bounded from above by employing Lemma 3.4:

$$\mathbb{E}_{\mathcal{A}} \left[ \sum_{t=1}^{T} \left\langle \widetilde{\mathbf{g}}^{(t)}, \boldsymbol{p}^{(t)} - C\boldsymbol{e}_{j^{\mathcal{A}}} \right\rangle \right] \leq \mathbb{E}_{\mathcal{A}} \left[ \frac{\eta \sum_{t=1}^{T} \left\| \widetilde{\mathbf{g}}^{(t)} \right\|_q^2}{2\alpha} + \frac{1}{\eta} B_{\Phi}(C\boldsymbol{e}_{j^{\mathcal{A}}}, \boldsymbol{p}^{(1)}) \right].$$

Similarly, denote $G := \bar{\gamma}^2 n^2 \|\boldsymbol{b}\|_q^2$ and $G_{\mathcal{A}} := \sum_{t=1}^{T} \|\mathbf{n}^{(t)}\|_q^2 / T$. Then $\mathbb{E}_{\mathcal{A}}\left[\sum_{t=1}^{T} \|\widetilde{\mathbf{g}}^{(t)}\|_q^2\right] \leq 2T \cdot \left(G + \sigma^2 \mathbb{E}\left[\|\boldsymbol{z}\|_q^2\right]\right)$ with $\boldsymbol{z}$ being a standard Gaussian vector. Plugging these values back, we get

$$\mathbb{E}_{\mathcal{A}}\left[\sum_{t=1}^{T}\left\langle\widetilde{\mathbf{g}}^{(t)}, \boldsymbol{p}^{(t)} - C\boldsymbol{e}_{j^{\mathcal{A}}}\right\rangle\right] \leq \frac{\eta T \cdot \left(G + \sigma^2 \mathbb{E}\left[\|\boldsymbol{z}\|_q^2\right]\right)}{\alpha} + \frac{1}{\eta}\mathbb{E}_{\mathcal{A}}\left[B_{\Phi}(C\boldsymbol{e}_{j^{\mathcal{A}}}, \boldsymbol{p}^{(1)})\right].$$

Lastly, using stepsize $\eta = \sqrt{\frac{\alpha B_{\Phi}(\boldsymbol{0}, \boldsymbol{p}^{(1)})}{T \cdot (G + \sigma^2 \mathbb{E}[\|\boldsymbol{z}\|_q^2])}}$ results in

$$\mathbb{E}_{\mathcal{A}}\left[\sum_{t=1}^{T}\left\langle\mathbf{g}^{(t)}, \boldsymbol{p}^{(t)} - C\boldsymbol{e}_{j^{\mathcal{A}}}\right\rangle\right] \leq \sqrt{\frac{T \cdot (G + \sigma^2 \mathbb{E}[\|\boldsymbol{z}\|_q^2])}{\alpha}} \cdot \frac{B_{\Phi}(\boldsymbol{0}, \boldsymbol{p}^{(1)}) + \mathbb{E}_{\mathcal{A}}\left[B_{\Phi}(C\boldsymbol{e}_{j^{\mathcal{A}}}, \boldsymbol{p}^{(1)})\right]}{\sqrt{B_{\Phi}(\boldsymbol{0}, \boldsymbol{p}^{(1)})}}$$

$$\leq \sqrt{\frac{T \cdot (G + \sigma^2 \mathbb{E}[\|\boldsymbol{z}\|_q^2])}{\alpha}} \cdot \underbrace{\frac{B_{\Phi}(\boldsymbol{0}, \boldsymbol{p}^{(1)}) + \max_{\boldsymbol{e} \in E_1} B_{\Phi}(C\boldsymbol{e}, \boldsymbol{p}^{(1)})}{\sqrt{B_{\Phi}(\boldsymbol{0}, \boldsymbol{p}^{(1)})}}}_{=:C_{\Phi,1}(\boldsymbol{p}^{(1)})}. \tag{16}$$

Comparing the lower bound (15) with the upper bound (16) and using $\sigma^2 = T c_{\varepsilon,\delta}$, we immediately obtain

$$T\mathbb{E}_{\mathcal{A}}\left[\|\boldsymbol{v}^{\mathcal{A}}\|_{\infty}\right] (C - \|\boldsymbol{p}^*\|_1) - CT\sqrt{c_{\varepsilon,\delta} \cdot 2\ln m} \leq \frac{C_{\Phi,1}(\boldsymbol{p}^{(1)})}{\sqrt{\alpha}} \cdot \left(\sqrt{TG} + T\sqrt{c_{\varepsilon,\delta}} \cdot \sqrt{\mathbb{E}\left[\|\boldsymbol{z}\|_q^2\right]}\right).$$

Rearranging the above inequality yields (assume $C > \|\boldsymbol{p}^*\|_1$)

$$\mathbb{E}_{\mathcal{A}}\left[\|\boldsymbol{v}^{\mathcal{A}}\|_{\infty}\right] \leq \frac{C_{\Phi,1}(\boldsymbol{p}^{(1)})}{\sqrt{\alpha} \cdot (C - \|\boldsymbol{p}^*\|_1)} \cdot \left(\sqrt{\frac{G}{T}} + \sqrt{c_{\varepsilon,\delta}} \cdot \sqrt{\mathbb{E}\left[\|\boldsymbol{z}\|_q^2\right]}\right) + \frac{C\sqrt{2\ln m}}{C - \|\boldsymbol{p}^*\|_1} \cdot \sqrt{c_{\varepsilon,\delta}}. \tag{17}$$

When $T \geq \frac{G}{c_{\varepsilon,\delta} \cdot \mathbb{E}\left[\|\boldsymbol{z}\|_q^2\right]}$, (17) implies

$$\mathbb{E}_{\mathcal{A}}\left[\|\boldsymbol{v}^{\mathcal{A}}\|_{\infty}\right] \leq \left(\frac{2\sqrt{\mathbb{E}[\|\boldsymbol{z}\|_q^2]} \cdot C_{\Phi,1}(\boldsymbol{p}^{(1)})}{\sqrt{\alpha} \cdot (C - \|\boldsymbol{p}^*\|_1)} + \frac{C\sqrt{2\ln m}}{C - \|\boldsymbol{p}^*\|_1}\right) \cdot \sqrt{c_{\varepsilon,\delta}}. \tag{18}$$

The inequality (18) gives an upper bound on the expected maximal constraint violation.

However, the above analysis holds only when constant $C$ satisfies (i) $C > \|\boldsymbol{p}^*\|_1$ and (ii) $C\boldsymbol{e}_{j^{\mathcal{A}}} \in \mathbb{R}_+^m \cap \mathcal{W}$ for any $j^{\mathcal{A}} \in [m]$. To this end, we can simply let $C = 2\|\boldsymbol{p}^*\|_1$, and replacing $C$ with $2\|\boldsymbol{p}^*\|_1$ gives the desired result:

$$\mathbb{E}_{\mathcal{A}}\left[\|\boldsymbol{v}^{\mathcal{A}}\|_{\infty}\right] \leq \left(\frac{2\sqrt{\mathbb{E}\left[\|\boldsymbol{z}\|_q^2\right]} \cdot C_{\Phi,1}(\boldsymbol{p}^{(1)})}{\sqrt{\alpha} \cdot \|\boldsymbol{p}^*\|_1} + 2\sqrt{2\ln m}\right) \cdot \sqrt{c_{\varepsilon,\delta}}. \tag{19}$$

$\square$

## A.5 Proof of Lemma 3.8

*Proof.* For any agent $i$, the objective value of its individual dual problem is always positive, i.e., $\max_{\boldsymbol{x}_i \in \mathcal{X}_i}\{u_i(\boldsymbol{x}_i) - \langle\boldsymbol{p}^*, \boldsymbol{a}_i(\boldsymbol{x}_i)\rangle\} \geq 0$ by Assumption 2.5 part 4. With this inequality, we are ready to show bounded shadow prices:

$$n\gamma\langle\boldsymbol{b}, \boldsymbol{p}^*\rangle + 0 \leq \langle n\gamma\boldsymbol{b}, \boldsymbol{p}^*\rangle + \max_{\boldsymbol{x} \in \mathcal{X}}\{\mathsf{F}(\boldsymbol{x}) - \langle\boldsymbol{p}^*, \boldsymbol{a}(\boldsymbol{x})\rangle\} = \underbrace{\mathsf{D}(\boldsymbol{p}^*) = \mathsf{F}(\boldsymbol{x}^*)}_{\text{(by strong duality)}} \leq n\bar{u}.$$

Dividing both sides by $n\gamma$ results in $\langle \boldsymbol{b}, \boldsymbol{p}^* \rangle \leq \frac{\bar{u}}{\gamma}$.

Similarly,

$$n\gamma \langle \boldsymbol{1}, \boldsymbol{p}^* \rangle \leq \frac{\langle n\gamma \boldsymbol{b}, \boldsymbol{p}^* \rangle + 0}{\min_j b_j} \leq \frac{\langle n\gamma \boldsymbol{b}, \boldsymbol{p}^* \rangle + \max_{\boldsymbol{x} \in \mathcal{X}} \{ \mathsf{F}(\boldsymbol{x}) - \langle \boldsymbol{p}^*, \boldsymbol{a}(\boldsymbol{x}) \rangle \}}{\min_j b_j}$$

$$= \underbrace{\frac{\mathsf{D}(\boldsymbol{p}^*)}{\min_j b_j} = \frac{\mathsf{F}(\boldsymbol{x}^*)}{\min_j b_j}}_{\text{(by strong duality)}} \leq \frac{n\bar{u}}{\min_j b_j},$$

which implies $\langle \boldsymbol{1}, \boldsymbol{p}^* \rangle \leq \frac{\bar{u}}{\gamma \min_j b_j}$.

$\square$

## A.6 Proof of Lemma 3.9

*Proof.* By the definition of strong convexity, it suffices to show

$$\langle \nabla \Phi(\boldsymbol{p}; \boldsymbol{\theta}) - \nabla \Phi(\boldsymbol{q}; \boldsymbol{\theta}), \boldsymbol{p} - \boldsymbol{q} \rangle \geq \frac{(\min_j \{\theta_j\})^2}{K} \|\boldsymbol{p} - \boldsymbol{q}\|_1^2, \quad \forall \boldsymbol{p}, \boldsymbol{q} \in \Delta_K^{\mathrm{int}}(\boldsymbol{\theta}). \tag{20}$$

According to the definition of parameterized negative entropy function, its gradient is $\nabla \Phi(\boldsymbol{p}; \boldsymbol{\theta})_j = \theta_j \ln (p_j \theta_j) + \theta_j, \forall j \in [m]$. By the well celebrated Hermite–Hadamard inequality $f(\frac{a+b}{2}) \leq \frac{1}{b-a} \int_a^b f(x)\, dx$, if we use convex function $f(t) = \frac{1}{t}, \forall t > 0$, we immediately have $\frac{2}{a+b} \leq \frac{\ln b - \ln a}{b-a}$, which implies

$$(\ln b - \ln a) \cdot (b - a) \geq \frac{2(b-a)^2}{b+a}, \quad \forall a, b \geq 0, \tag{21}$$

where the case $a = 0$ or $b = 0$, or both are obtained by continuity and the conventions that $\frac{0}{0} = 0$ and $0 \ln 0 = 0$. If we set $b = p_j \theta_j$ and $a = q_j \theta_j$ for (21), we can get

$$(\ln (p_j \theta_j) - \ln (q_j \theta_j)) \cdot \theta_j (p_j - q_j) \geq \frac{2\theta_j (p_j - q_j)^2}{p_j + q_j}, \quad \forall j \in [m]. \tag{22}$$

Summing over $j \in [m]$, we have

$$\text{LHS of (20)} = \sum_{j=1}^m (\ln (p_j \theta_j) - \ln (q_j \theta_j)) \cdot \theta_j (p_j - q_j) \geq \sum_{j=1}^m \frac{2\theta_j (p_j - q_j)^2}{p_j + q_j}. \quad \text{(by (22))}$$

It remains to further lower bound the summation on the r.h.s. We follow the proof idea of Lemma 2 in [BLM23] to complete our proof:

$$\sum_{j=1}^m \frac{2\theta_j (p_j - q_j)^2}{p_j + q_j} = 2 \langle \boldsymbol{\theta}, \boldsymbol{p} + \boldsymbol{q} \rangle \cdot \sum_{j=1}^m \frac{\theta_j (p_j + q_j)}{\langle \boldsymbol{\theta}, \boldsymbol{p} + \boldsymbol{q} \rangle} \cdot \frac{|p_j - q_j|^2}{(p_j + q_j)^2}$$

$$\geq 2 \langle \boldsymbol{\theta}, \boldsymbol{p} + \boldsymbol{q} \rangle \cdot \left( \sum_{j=1}^m \frac{\theta_j (p_j + q_j)}{\langle \boldsymbol{\theta}, \boldsymbol{p} + \boldsymbol{q} \rangle} \cdot \frac{|p_j - q_j|}{p_j + q_j} \right)^2 \quad \text{(by Jensen's inequality)}$$

$$= \frac{2}{\langle \boldsymbol{\theta}, \boldsymbol{p} + \boldsymbol{q} \rangle} \cdot \left( \sum_{j=1}^m \theta_j |p_j - q_j| \right)^2$$

$$\geq \frac{1}{K} \cdot \left( \sum_{j=1}^m \theta_j |p_j - q_j| \right)^2 \quad \text{(since } \boldsymbol{p}, \boldsymbol{q} \in \Delta_K^{\mathrm{int}}(\boldsymbol{\theta}))$$

$$\geq \frac{(\min_j \{\theta_j\})^2}{K} \cdot \|\boldsymbol{p} - \boldsymbol{q}\|_1^2.$$

$\square$

## A.7 Proof of Theorem 3.10

*Proof.* The proof follows the general proof of Theorem 3.5 and 3.6 for any potential function $\Phi$. It remains to properly modify results for the $\Phi$ chosen. When the potential function is the parameterized negative entropy $\Phi := \Phi(\boldsymbol{w}; \boldsymbol{b}) = \sum_{j=1}^{m} w_j b_j \ln(w_j b_j)$ defined on $\Delta_K^{\text{int}}(\boldsymbol{b}) := \{\boldsymbol{w} > \boldsymbol{0} : \langle \boldsymbol{b}, \boldsymbol{w} \rangle \leq K\}$ with $K = \frac{2\bar{u}}{\gamma}$, the conjugate index $q = \infty$, strong convexity parameter $\alpha = \underline{b}^2/K$. Moreover, direct calculation gives the Bregman divergence $B_\Phi(\boldsymbol{0}, \boldsymbol{p}^{(1)}) = \langle \boldsymbol{b}, \boldsymbol{p}^{(1)} \rangle$. Because we choose $p_j^{(1)} = K/(mb_j), \forall j \in [m]$, we thus have $B_\Phi(\boldsymbol{0}, \boldsymbol{p}^{(1)}) = K$.

**Utility Guarantee:** It is evident that $C_\Phi(\boldsymbol{p}^{(1)}) := \sqrt{B_\Phi(\boldsymbol{0}, \boldsymbol{p}^{(1)})/\alpha} = \sqrt{K/(\underline{b}^2/K)} = K/\underline{b}$. Therefore, the utility upper bound (7) becomes

$$\mathsf{F}(\boldsymbol{x}^*) - \mathbb{E}_{\mathcal{A}}\left[\mathsf{F}(\boldsymbol{x}^{\mathcal{A}})\right] = 4C_\Phi(\boldsymbol{p}^{(1)}) \cdot \sqrt{\mathbb{E}\left[\|\boldsymbol{z}\|_\infty^2\right]} \cdot \sqrt{c_{\varepsilon,\delta}}$$

$$\leq 4\frac{K}{\underline{b}} \cdot \sqrt{2\ln(2m)} \cdot \sqrt{c_{\varepsilon,\delta}} = \frac{8\sqrt{2}\bar{u} \cdot \sqrt{\ln(2m)}}{\gamma\underline{b}} \cdot \sqrt{c_{\varepsilon,\delta}}.$$

**Feasibility Guarantee:** We can work from (19) and first figure out $C_{\Phi,1}(\boldsymbol{p}^{(1)})$. Direct calculation gives $B_\Phi(C\boldsymbol{e}_j, \boldsymbol{p}^{(1)}) = B_\Phi(\boldsymbol{0}, \boldsymbol{p}^{(1)}) + Cb_j \cdot \left(\ln(C/p_j^{(1)}) - 1\right), \forall j \in [m]$. Plugging $C = 2\|\boldsymbol{p}^*\|_1$, we can upper bound $C_{\Phi,1}(\boldsymbol{p}^{(1)})$ as:

$$C_{\Phi,1}(\boldsymbol{p}^{(1)}) := \frac{B_\Phi(\boldsymbol{0}, \boldsymbol{p}^{(1)}) + \max_j B_\Phi(C\boldsymbol{e}_j, \boldsymbol{p}^{(1)})}{\sqrt{B_\Phi(\boldsymbol{0}, \boldsymbol{p}^{(1)})}}$$

$$= 2\sqrt{K} + \frac{2\|\boldsymbol{p}^*\|_1 \max_j\left\{b_j \cdot \left(\ln\left(\frac{2\|\boldsymbol{p}^*\|_1 mb_j}{K}\right) - 1\right), 0\right\}}{\sqrt{K}} \qquad \text{(since } p_j^{(1)} = K/(mb_j))$$

$$\leq 2\sqrt{K} + \sqrt{K} \cdot \max_j \{b_j \cdot (\ln(mb_j) - 1), 0\} \qquad (K \geq 2\|\boldsymbol{p}^*\|_1, \text{ Lemma 3.8})$$

$$\leq 2\sqrt{K} + \sqrt{K}\bar{b} \cdot (\ln(m\bar{b}) - 1)^+.$$

Substitute $C_{\Phi,1}(\boldsymbol{p}^{(1)})$ back into (19) with $K = \frac{2\bar{u}}{\gamma}$, we get

$$\mathbb{E}_{\mathcal{A}}\left[\|\boldsymbol{v}^{\mathcal{A}}\|_\infty\right] \leq \frac{4\bar{u}\sqrt{2\ln(2m)} \cdot \left[2 + \bar{b} \cdot (\ln(m\bar{b}) - 1)^+\right]}{\gamma\underline{b} \cdot \|\boldsymbol{p}^*\|_1} \cdot \sqrt{c_{\varepsilon,\delta}}.$$

$\square$

# B  Proofs for Section 4

## B.1  Proof of self post-processing Lemma 4.1 for JDP

*Proof.* The proof is for a deterministic function $f$. The result on randomized functions follows immediately as any randomized mapping can be decomposed into a convex combination of deterministic functions. The proof here largely follows the same idea for the proof of Billboard Lemma. For any $\boldsymbol{x} \in \mathcal{X}^n$, let $\boldsymbol{f}(\boldsymbol{x}_{-k}, \mathcal{D}_{-k}) := (f(x_1, z_1), \ldots, f(x_{k-1}, z_{k-1}), f(x_{k+1}, z_{k+1}), \ldots, f(x_n, z_n))$. For any subset $\mathcal{S} \subseteq \mathcal{Y}^{n-1}$, let $\boldsymbol{f}^{-1}(\mathcal{S}; \mathcal{D}_{-k}) := \{\boldsymbol{x}_{-k} \in \mathcal{X}^{n-1} : \boldsymbol{f}(\boldsymbol{x}_{-k}, \mathcal{D}_{-k}) \in \mathcal{S}\}$ be the preimage of $\mathcal{S}$ with respect to the mapping $\boldsymbol{f}(\cdot, \mathcal{D}_{-k})$. Then, we can show that

$$\Pr\left[\boldsymbol{f}(\mathcal{M}(\mathcal{D})_{-k}, \mathcal{D}_{-k}) \in \mathcal{S}\right] = \Pr\left[\mathcal{M}(\mathcal{D})_{-k} \in \boldsymbol{f}^{-1}(\mathcal{S}; \mathcal{D}_{-k})\right]$$

$$\leq e^\varepsilon \cdot \Pr\left[\mathcal{M}(\mathcal{D}')_{-k} \in \boldsymbol{f}^{-1}(\mathcal{S}; \mathcal{D}_{-k})\right] + \delta \quad (\mathcal{M}(\cdot) \text{ is } (\varepsilon, \delta)\text{-JDP})$$

$$= e^\varepsilon \cdot \Pr\left[\boldsymbol{f}(\mathcal{M}(\mathcal{D}')_{-k}, \mathcal{D}_{-k}) \in \mathcal{S}\right] + \delta$$

$$= e^\varepsilon \cdot \Pr\left[\boldsymbol{f}(\mathcal{M}(\mathcal{D}')_{-k}, \mathcal{D}'_{-k}) \in \mathcal{S}\right] + \delta, \qquad (\mathcal{D}_{-k} = \mathcal{D}'_{-k})$$

which confirms JDP guarantee after self post-processing.

$\square$

## B.2 Proof of post-processing Lemma 4.2 for JDP

*Proof.* Similarly, the proof is for a deterministic function $f$. Let $f^{-1}(\mathcal{S})$ be the preimage of $\mathcal{S}$. For any $\mathcal{S} \subseteq \mathcal{Y}$:

$$
\begin{aligned}
\Pr\left[f(\mathcal{M}(\mathcal{D})_{-k}) \in \mathcal{S}\right] &= \Pr\left[\mathcal{M}(\mathcal{D})_{-k} \in f^{-1}(\mathcal{S})\right] \\
&\leq e^{\varepsilon} \cdot \Pr\left[\mathcal{M}(\mathcal{D}')_{-k} \in f^{-1}(\mathcal{S})\right] + \delta \qquad (\mathcal{M}(\cdot) \text{ is } (\varepsilon, \delta)\text{-JDP}) \\
&= e^{\varepsilon} \cdot \Pr\left[f(\mathcal{M}(\mathcal{D}')_{-k}) \in \mathcal{S}\right] + \delta.
\end{aligned}
$$

$\square$

## B.3 Proof of Theorem 4.3

The proof consists of several steps: (i) construct a "hard" distribution of datasets; (ii) identify necessary conditions on the JDP algorithm $\mathcal{A}$ with the help of Lemma 4.1 and 4.2; (iii) reformulate the minimax lower bound into a better form that aligns these necessary conditions; (iv) combine all steps together. Without loss of generality, we assume $b = 1$.

**Step 1: Construct a "hard" distribution of datasets.**

Let $\mathrm{Id}(\cdot)$ be the identity function, and let $\mathbf{1}$ and $\mathbf{0}$ be vectors of 1's and 0's whose lengths are $m$. Define three types of requests:

$$
\begin{aligned}
z &:= (\boldsymbol{x} \mapsto \langle \mathbf{1}, \boldsymbol{x} \rangle, \ \mathrm{Id}(\cdot), \ \{\mathbf{1}x : x \in [0,1]\}); \\
z^1 &:= (\boldsymbol{x} \mapsto \langle \mathbf{1}, \boldsymbol{x} \rangle, \ \mathrm{Id}(\cdot), \ \{\mathbf{1}x : x = \min\{1, n\gamma\}\}); \\
z^0 &:= (\boldsymbol{x} \mapsto \langle \mathbf{1}, \boldsymbol{x} \rangle, \ \mathrm{Id}(\cdot), \ \{\mathbf{1}x : x = 0\}).
\end{aligned}
$$

All three types of requests are the same in their utility and consumption functions, both taking linear forms. It is also easy to verify that all three types of requests satisfy Assumptions 2.5 (part 1 and 2). In what follows, we construct datasets that satisfy Assumption 2.5 (part 3 and 4) and Assumption 3.7 as well.

Based on these requests, we construct a pair of neighboring datasets that differ in agent $n$ only:

$$
\mathcal{D}_0 := (\underbrace{z, \ldots, z}_{n-1 \text{ requests}}, \ z^0); \qquad \mathcal{D}_1 := (\underbrace{z, \ldots, z}_{n-1 \text{ requests}}, \ z^1).
$$

It is easy to verify that part 3 of Assumption 2.5 is also met, i.e., the primal problem is always feasible, and the optimal shadow price is $\boldsymbol{p}^* = \mathbf{1}$, which are non-zeros. The result $\boldsymbol{p}^* = \mathbf{1}$ is by noticing that one unit increase in available resource will increase the objective value of the primal problem by 1. Since $\boldsymbol{p}^* = \mathbf{1}$, it immediate to see that part 4 of Assumption 2.5 also holds, because now the function $u_i(\boldsymbol{x}_i) + \langle \boldsymbol{p}^*, -\boldsymbol{a}_i(\boldsymbol{x}_i) \rangle$ becomes $\langle \mathbf{1}, \boldsymbol{x}_i \rangle + \langle \mathbf{1}, -\boldsymbol{x}_i \rangle$, which is always non-negative for any $\boldsymbol{x}_i \in \mathcal{X}_i$. Moreover, the constraint to be dualized is a linear constraint; thus Assumption 3.7 strong duality holds.

Lastly, let dataset $\mathcal{D}$ be drawn from $\mathrm{Unif}\{\mathcal{D}_0, \mathcal{D}_1\}$, i.e. uniformly distributed over these two datasets.

**Step 2: Identify necessary conditions on JDP algorithm $\mathcal{A}$**

Let $\boldsymbol{x}^{\mathcal{A}}(\mathcal{D}) := (\boldsymbol{x}_1^{\mathcal{A}}(\mathcal{D}), \ldots, \boldsymbol{x}_n^{\mathcal{A}}(\mathcal{D}))$ be the output of $(\varepsilon, \delta)$-JDP algorithm $\mathcal{A}$. The self post-processing Lemma 4.1 says that if we process each $\boldsymbol{x}_i^{\mathcal{A}}$ with $i$'s own data, the resulting output is still JDP. That implies $(\mathrm{Id}(\boldsymbol{x}_1^{\mathcal{A}}), \ldots, \mathrm{Id}(\boldsymbol{x}_n^{\mathcal{A}}))$ should also be $(\varepsilon, \delta)$-JDP. Moreover, the post-processing Lemma 4.2 indicates that the aggregated consumption, excluding agent $n$'s, should be indistinguishable even if agent $n$'s data is changed, i.e.,

$$
\Pr\left[\sum_{i \neq n} \mathrm{Id}(\boldsymbol{x}_i^{\mathcal{A}}(\mathcal{D})) \in \mathcal{S}\right] \leq e^{\varepsilon} \Pr\left[\sum_{i \neq n} \mathrm{Id}(\boldsymbol{x}_i^{\mathcal{A}}(\mathcal{D}')) \in \mathcal{S}\right] + \delta, \quad \forall \mathcal{S} \subseteq [\mathbf{0}, n\gamma\mathbf{1}].
$$

Instantiating the above inequality with $\mathcal{D} = \mathcal{D}_0$, $\mathcal{D}' = \mathcal{D}_1$, and $\mathcal{S} = (n\gamma\mathbf{1} - c\mathbf{1}, n\gamma\mathbf{1}]$ with $c := \min\{1, n\gamma\}$ leads to

$$
\Pr\left[\sum_{i \neq n} \boldsymbol{x}_i^{\mathcal{A}}(\mathcal{D}_0) \in (n\gamma\mathbf{1} - c\mathbf{1}, n\gamma\mathbf{1}]\right] \leq e^{\varepsilon} \Pr\left[\sum_{i \neq n} \boldsymbol{x}_i^{\mathcal{A}}(\mathcal{D}_1) \in (n\gamma\mathbf{1} - c\mathbf{1}, n\gamma\mathbf{1}]\right] + \delta.
$$

We can simplify the preceding inequality by changing decision variables from bold format $\boldsymbol{x}_i$ to normal format $x_i$, because by construction of requests $z, z^1, z^0$, each decision vector $\boldsymbol{x}_i := \mathbf{1}x_i$ is fully determined by a single scalar $x_i$. This leads to the following equivalent inequality,

$$\Pr\left[\sum_{i \neq n} x_i^{\mathcal{A}}(\mathcal{D}_0) \in (n\gamma - c, n\gamma]\right] \leq e^{\varepsilon} \Pr\left[\sum_{i \neq n} \boldsymbol{x}_i^{\mathcal{A}}(\mathcal{D}_1) \in (n\gamma - c, n\gamma]\right] + \delta. \qquad (23)$$

Moreover, when dataset is $\mathcal{D}_1$, agent $n$ should always be assigned $c$ unit, forcing resource available for others to be at most $n\gamma - c$. Thus, if $\mathcal{A}$ is an $(\varepsilon, \delta)$-JDP algorithm outputting a feasible allocation with probability 1, then $\sum_{i \neq n} x_i^{\mathcal{A}}(\mathcal{D}_1) \leq n\gamma - c$ w.p. 1, and (23) becomes:

$$\Pr\left[\sum_{i \neq n} x_i^{\mathcal{A}}(\mathcal{D}_0) \in (n\gamma - c, n\gamma]\right] \leq e^{\varepsilon} \cdot 0 + \delta = \delta. \qquad (24)$$

Therefore, we get a necessary condition (24) for any $\mathcal{A}$ being $(\varepsilon, \delta)$-JDP: when dataset is $\mathcal{D}_0$, Algorithm $\mathcal{A}$ should not assign many resources to other agents, and only with a small probability $\delta$, other agents can get at least $n\gamma - c$ resources.

### Step 3: Reformulate the minimax bound

Let $\Delta \in [0, n\gamma]$ be a constant to be determined later, and let $S_{\Delta}^+ := (n\gamma - \Delta, n\gamma]$, $S_{\Delta}^- := [0, n\gamma - \Delta]$ be two disjoint regions partitioning the interval $[0, n\gamma]$. Let $\mathcal{I} \in \{0, 1\}^m$ be an indicator so that 1 in $j$'s position implies $(\sum_{i=1}^n \boldsymbol{x}_i^{\mathcal{A}})_j \in S_{\Delta}^+$; otherwise, in $S_{\Delta}^-$. Let event $E_{\mathcal{I}} := \{(\boldsymbol{x}_1, \ldots, \boldsymbol{x}_n) : \mathbb{1}\{(\sum_{i=1}^n \boldsymbol{x}_i)_j \in S_{\Delta}^+\} = \mathcal{I}_j, \forall j \in [m]\}$. With these notations and the law of total probability, we first rewrite the expression of Algorithm $\mathcal{A}$'s utility when dataset is $\mathcal{D}_0$ as:

$$\mathbb{E}_{\mathcal{A}}\left[\mathsf{F}(\boldsymbol{x}^{\mathcal{A}}(\mathcal{D}_0))\right] = \sum_{\mathcal{I} \in \{0,1\}^m} \mathbb{E}_{\mathcal{A}}\left[\sum_{i=1}^n \langle \mathbf{1}, \boldsymbol{x}_i^{\mathcal{A}}(\mathcal{D}_0) \rangle \;\Big|\; \boldsymbol{x}^{\mathcal{A}}(\mathcal{D}_0) \in E_{\mathcal{I}}\right] \cdot \Pr\left[\boldsymbol{x}^{\mathcal{A}}(\mathcal{D}_0) \in E_{\mathcal{I}}\right]$$

$$= \sum_{\mathcal{I} \in \{0,1\}^m} \mathbb{E}_{\mathcal{A}}\left[\sum_{i=1}^n \sum_{j=1}^m \left(\boldsymbol{x}_i^{\mathcal{A}}(\mathcal{D}_0)\right)_j \;\Big|\; \boldsymbol{x}^{\mathcal{A}}(\mathcal{D}_0) \in E_{\mathcal{I}}\right] \cdot \Pr\left[\boldsymbol{x}^{\mathcal{A}}(\mathcal{D}_0) \in E_{\mathcal{I}}\right]$$

$$= \sum_{\mathcal{I} \in \{0,1\}^m} \mathbb{E}_{\mathcal{A}}\left[\sum_{j=1}^m \left(\sum_{i=1}^n \boldsymbol{x}_i^{\mathcal{A}}(\mathcal{D}_0)\right)_j \;\Big|\; \boldsymbol{x}^{\mathcal{A}}(\mathcal{D}_0) \in E_{\mathcal{I}}\right] \cdot \Pr\left[\boldsymbol{x}^{\mathcal{A}}(\mathcal{D}_0) \in E_{\mathcal{I}}\right].$$

We observe that when event $E_{\mathcal{I}}$ happens, the resource consumed $\left(\sum_{i=1}^n \boldsymbol{x}_i^{\mathcal{A}}(\mathcal{D}_0)\right)_j$ is no more than either $n\gamma - \Delta$ or $n\gamma$, depending on $\mathcal{I}_j$. Hence, we can reach an upper bound expressed by $\mathcal{I}_j$:

$$\mathbb{E}_{\mathcal{A}}\left[\mathsf{F}(\boldsymbol{x}^{\mathcal{A}}(\mathcal{D}_0))\right] \leq \sum_{\mathcal{I} \in \{0,1\}^m} \mathbb{E}_{\mathcal{A}}\left[\sum_{j=1}^m [(n\gamma - \Delta)\mathbb{1}\{\mathcal{I}_j = 0\} + n\gamma \mathbb{1}\{\mathcal{I}_j = 1\}] \;\Big|\; E_{\mathcal{I}}\right] \cdot \Pr\left[\boldsymbol{x}^{\mathcal{A}}(\mathcal{D}_0) \in E_{\mathcal{I}}\right]$$

$$= \sum_{\mathcal{I} \in \{0,1\}^m} \mathbb{E}_{\mathcal{A}}\left[mn\gamma - \sum_{j=1}^m \Delta \mathbb{1}\{\mathcal{I}_j = 0\} \;\Big|\; E_{\mathcal{I}}\right] \cdot \Pr\left[\boldsymbol{x}^{\mathcal{A}}(\mathcal{D}_0) \in E_{\mathcal{I}}\right]$$

$$= mn\gamma - \Delta \sum_{j=1}^m \sum_{\mathcal{I} \in \{0,1\}^m} \mathbb{E}_{\mathcal{A}}\left[\mathbb{1}\{\mathcal{I}_j = 0\} \;\Big|\; E_{\mathcal{I}}\right] \Pr\left[\boldsymbol{x}^{\mathcal{A}}(\mathcal{D}_0) \in E_{\mathcal{I}}\right]$$

$$= mn\gamma - \Delta \sum_{j=1}^m \mathbb{E}_{\mathcal{A}}\left[\mathbb{1}\{\mathcal{I}_j = 0\}\right]$$

$$= mn\gamma - \Delta \sum_{j=1}^m \Pr\left[\left(\sum_{i=1}^n \boldsymbol{x}_i^{\mathcal{A}}(\mathcal{D}_0)\right)_j \in S_{\Delta}^-\right].$$

Due to the construction of $\mathcal{D}_0$ that $\boldsymbol{x}_i^{\mathcal{A}} := \mathbf{1}x_i^{\mathcal{A}}$, the probability terms in the preceding inequality should be the same among all $j = 1, \ldots, m$; thus, we have

$$\mathbb{E}_{\mathcal{A}}\left[\mathsf{F}(\boldsymbol{x}^{\mathcal{A}}(\mathcal{D}_0))\right] \leq mn\gamma - m\Delta \cdot \Pr\left[\sum_{i=1}^{n} x_i^{\mathcal{A}}(\mathcal{D}_0) \in S_\Delta^-\right].$$

Moreover, with $\mathcal{D}_0$, the allocation to agent $n$ should be $x_n^{\mathcal{A}}(\mathcal{D}_0) = 0$, modifying the preceding inequality into

$$\mathbb{E}_{\mathcal{A}}\left[\mathsf{F}(\boldsymbol{x}^{\mathcal{A}}(\mathcal{D}_0))\right] \leq mn\gamma - m\Delta \cdot \Pr\left[\sum_{i \neq n} x_i^{\mathcal{A}}(\mathcal{D}_0) \in S_\Delta^-\right]. \tag{25}$$

We note that with either dataset $\mathcal{D}_0$ or $\mathcal{D}_1$, the optimal utility obtained in the non-private setting is always $mn\gamma$, i.e., $\mathsf{F}(\boldsymbol{x}^*(\mathcal{D}_0)) = \mathsf{F}(\boldsymbol{x}^*(\mathcal{D}_1)) = mn\gamma$. As a result, the minimax regret is lower bounded as follows:

$$\inf_{\mathcal{A} \text{ is } (\varepsilon,\delta)\text{-JDP}} \sup_{\mathcal{D}} \left\{\mathsf{F}(\boldsymbol{x}^*(\mathcal{D})) - \mathbb{E}_{\mathcal{A}}\left[\mathsf{F}(\boldsymbol{x}^{\mathcal{A}}(\mathcal{D}))\right]\right\} \geq \inf_{\mathcal{A} \text{ is } (\varepsilon,\delta)\text{-JDP}} \mathbb{E}_{\mathcal{D} \sim \mathrm{Unif}\{\mathcal{D}_0,\mathcal{D}_1\}}\left[\mathsf{F}(\boldsymbol{x}^*(\mathcal{D})) - \mathbb{E}_{\mathcal{A}}\left[\mathsf{F}(\boldsymbol{x}^{\mathcal{A}}(\mathcal{D}))\right]\right]$$

$$\geq \frac{1}{2} \cdot \inf_{\mathcal{A} \text{ is } (\varepsilon,\delta)\text{-JDP}} \left\{mn\gamma - \left(mn\gamma - m\Delta \cdot \Pr\left[\sum_{i \neq n} x_i^{\mathcal{A}}(\mathcal{D}_0) \in S_\Delta^-\right]\right)\right\}$$

$$= \frac{m\Delta}{2} \cdot \inf_{\mathcal{A} \text{ is } (\varepsilon,\delta)\text{-JDP}} \Pr\left[\sum_{i \neq n} x_i^{\mathcal{A}}(\mathcal{D}_0) \in [0, n\gamma - \Delta]\right], \tag{26}$$

where the second inequality is by considering $\mathcal{D}_0$ and plugging in (25).

**Step 4: Combine all together**

Suppose $\varepsilon \geq \max\{1, 1/(n\gamma)\}$. If we set $\Delta = \frac{1}{\varepsilon}$, the r.h.s. of (26) becomes $\frac{m}{2\varepsilon} \cdot \inf_{\mathcal{A}} \Pr\left[\sum_{i \neq n} x_i^{\mathcal{A}}(\mathcal{D}_0) \in [0, n\gamma - 1/\varepsilon]\right]$. Since $1/\varepsilon \leq \min\{1, n\gamma\} = c$, it follows that

$$\Pr\left[\sum_{i \neq n} x_i^{\mathcal{A}}(\mathcal{D}_0) \in [0, n\gamma - 1/\varepsilon]\right] \geq \Pr\left[\sum_{i \neq n} x_i^{\mathcal{A}}(\mathcal{D}_0) \in [0, n\gamma - c]\right]$$

$$= 1 - \Pr\left[\sum_{i \neq n} x_i^{\mathcal{A}}(\mathcal{D}_0) \in (n\gamma - c, n\gamma]\right]$$

$$\geq 1 - \delta \geq \frac{1}{2}. \qquad\qquad \text{(by (24) and } \delta \in (0, 1/2))$$

Plugging the preceding inequality back into (26) immediately leads to

$$\inf_{\mathcal{A} \text{ is } (\varepsilon,\delta)\text{-JDP}} \sup_{\mathcal{D}} \left\{\mathsf{F}(\boldsymbol{x}^*(\mathcal{D})) - \mathbb{E}_{\mathcal{A}}\left[\mathsf{F}(\boldsymbol{x}^{\mathcal{A}}(\mathcal{D}))\right]\right\} \geq \frac{m}{4\varepsilon},$$

which completes the proof.

### B.4 Discussion on lower bounds for general $\varepsilon$

The lower bound in Theorem 4.3 only holds for $\varepsilon \geq \max\{1, 1/(n\gamma)\}$. Intuitively, this is because we only use feasibility of $\mathcal{A}$ to prove the lower bound: "we *reserve* some resource exclusively for agent $k$." The optimality condition of $\mathcal{A}$ in the minimax expression $\inf_{\mathcal{A}} \sup_{\mathcal{D}}$ is not used. Noticing this oversight, we provide a lower bounding problem that fills this gap.

We start with expressing DP in terms of hockey-stick divergence. Suppose $\mu$ and $\mu'$ are two probability measures that are absolutely continuous with respect to each other. Let $H_{e^\varepsilon}(X, Y) := \int_{z \in \mathcal{Z}} \left(\frac{d\mu}{d\mu'}(z) - e^\varepsilon\right)^+ d\mu'(z)$ be the hockey-stick divergence between two random variables $X, Y$.

Let $\mathcal{M}$ be an $(\varepsilon, \delta)$-DP mechanism, and let $\mathcal{M}(\mathcal{D})_{-k} := \Pi_{-k}\mathcal{M}(\mathcal{D})$ be the random vector of $n-1$ elements projected to others than agent $k$. Here are some relationships among indistinguishability, DP, and JDP.

- $(\varepsilon, \delta)$-**indistinguishability**: if $H_{e^\varepsilon}(X, Y) \leq \delta$, then $X$ is said to be $(\varepsilon, \delta)$-indistinguishable from $Y$;

- $(\varepsilon, \delta)$-**DP**: for a mechanism $\mathcal{M}$, if $\sup_{\mathcal{D} \sim \mathcal{D}'} H_{e^\varepsilon}(\mathcal{M}(\mathcal{D}), \mathcal{M}(\mathcal{D}')) \leq \delta$, we say $\mathcal{M}$ is $(\varepsilon, \delta)$-DP [BBG18];

- $(\varepsilon, \delta)$-**JDP**: for a mechanism $\mathcal{M}$, if $\sup_{\mathcal{D} \sim \mathcal{D}', k} H_{e^\varepsilon}(\mathcal{M}(\mathcal{D})_{-k}, \mathcal{M}(\mathcal{D}')_{-k}) \leq \delta$, we say $\mathcal{M}$ is $(\varepsilon, \delta)$-JDP.

It is self-evident from these relationships that DP requires indistinguishability between outputs for any pair of neighboring datasets, while JDP requires indistinguishability between projected outputs without agent $k$ for any pair of neighboring datasets.

To obtain another lower bound exploiting optimality of $\mathcal{A}$, we can follow the idea and notation in the proof of Theorem 4.3, but consider different neighboring datasets:

$$\mathcal{D}^{1-} := (z^0, \underbrace{z, \ldots, z}_{\text{n-1 requests}}), \quad \mathcal{D}^{1+} := (z^1, \underbrace{z, \ldots, z}_{\text{n-1 requests}}), \quad \mathcal{D}^{1+, 2+} := (z^1, z^1, \underbrace{z, \ldots, z}_{\text{n-2 requests}}).$$

It is obvious $\mathcal{D}^{1-} \sim \mathcal{D}^{1+}$, and $\mathcal{D}^{1+} \sim \mathcal{D}^{1+, 2+}$ are pairs of neighboring datasets. Let $S_{-k}(\boldsymbol{x}^{\mathcal{A}}) := \sum_{i \neq k} x_i^{\mathcal{A}}$ be the total consumption excluding agent $k$. By the definition of $(\varepsilon, \delta)$-JDP, any JDP algorithm $\mathcal{A}$ should satisfy following necessary constraints:

$$H_{e^\varepsilon}(S_{-1}(\boldsymbol{x}^{\mathcal{A}}(\mathcal{D}^{1-})), S_{-1}(\boldsymbol{x}^{\mathcal{A}}(\mathcal{D}^{1+}))) \leq \delta; \tag{27}$$

$$H_{e^\varepsilon}(S_{-1}(\boldsymbol{x}^{\mathcal{A}}(\mathcal{D}^{1+})), S_{-1}(\boldsymbol{x}^{\mathcal{A}}(\mathcal{D}^{1+, 2+}))) \leq \delta; \tag{28}$$

$$H_{e^\varepsilon}(S_{-2}(\boldsymbol{x}^{\mathcal{A}}(\mathcal{D}^{1+, 2+})), S_{-2}(\boldsymbol{x}^{\mathcal{A}}(\mathcal{D}^{1+}))) \leq \delta; \tag{29}$$

$$H_{e^\varepsilon}(S_{-2}(\boldsymbol{x}^{\mathcal{A}}(\mathcal{D}^{1+})), S_{-2}(\boldsymbol{x}^{\mathcal{A}}(\mathcal{D}^{1-}))) \leq \delta. \tag{30}$$

The variable $\boldsymbol{x}^{\mathcal{A}}(\mathcal{D}^{1+, 2+})$ in both constraints (28) and (29) should be the same, and thus links both constraints. Moreover, the variable $\boldsymbol{x}^{\mathcal{A}}(\mathcal{D}^{1-})$ in constraints (27) and (30) are the same. Therefore, the four constraints form a loop restricting each other.

The four constraints (27)-(30) are necessary conditions for any JDP algorithm. Furthermore, the dataset $\mathcal{D}^{1-}$ here is essentially the same as $\mathcal{D}_0$ in the proof of Theorem 4.3; thus all analysis for $\mathcal{D}_0$ can be borrowed. Consequently, following the same analysis for (26), we get a lower bound expressed by an inf problem:

$$\inf_{\mathcal{A} \text{ is } (\varepsilon, \delta)\text{-JDP}} \sup_{\mathcal{D}} \left\{ \mathsf{F}(\boldsymbol{x}^*(\mathcal{D})) - \mathbb{E}_{\mathcal{A}} \left[ \mathsf{F}(\boldsymbol{x}^{\mathcal{A}}(\mathcal{D})) \right] \right\} \geq \frac{m}{2\varepsilon} \cdot \inf_{\mathcal{A} \text{ s.t. (27)-(30)}} \Pr \left[ S_{-1}(\boldsymbol{x}^{\mathcal{A}}(\mathcal{D}^{1-})) \in [0, n\gamma - \frac{1}{\varepsilon}] \right].$$

We believe analyzing the inf problem with tools in [BBG18], such as couplings, is promising. After this base case, one can easily extend the set of constraints (27)-(30) to take account of more loops.

## C   Additional results of numerical experiments

All experiments were run on a PC with an AMD 3700X CPU, 16GB memory; no GPU was used. The algorithm was implemented in Python 3.11, and optimization was solved by scipy.optimize.minimize method. Source code is available in supplementary materials. Datasets are publicly available as discussed in the main text.

### C.1   Workforce scheduling

When running the algorithm, the only parameter different from Theorem 3.10 is $K := 1.1\bar{u}/(\gamma\underline{b})$, where Theorem 3.10 suggests $K := 2\bar{u}/(\gamma\underline{b})$. The constant factor is modified from 2 to 1.1, as a result of hyperparameter tuning. We discuss the tuning and its impact later.

We first report optimality gaps in Table 4. It is clear that our algorithm, equipped with either potential

Table 4: Optimality gap $(\mathsf{F}(\boldsymbol{x}^*) - \mathsf{F}(\boldsymbol{x}^{\mathcal{A}}))/\mathsf{F}(\boldsymbol{x}^*) \times 100\%$. mean$\pm$sd

| Algorithms | MD_ne | MD_l2 | [HHRW16], Algo 1 |
|---|---|---|---|
| $\varepsilon = 1$ | **2.1**$\pm$2.4 | 9.1$\pm$3.8 | 21.5$\pm$1.2 |
| $\varepsilon = 2$ | **2.8**$\pm$2.1 | 7.4$\pm$4.1 | 21.8$\pm$1.2 |
| $\varepsilon = 5$ | **2.1**$\pm$1.9 | 6.6$\pm$4.7 | 22.4$\pm$1.0 |
| $\varepsilon = 10$ | **2.8**$\pm$2.1 | 5.3$\pm$4.1 | 23.3$\pm$1.0 |
| $\varepsilon = 20$ | **2.8**$\pm$2.4 | 4.2$\pm$2.9 | 24.2$\pm$1.0 |

*Notes.* We run our algorithm $\mathcal{A}$ Noisy Dual MD with two potential functions: negative entropy (abbr. ne) and squared $\ell_2$-norm function (abbr. $\ell_2$). Settings are the same as in Figure 1. **bold**=better

functions, always outperform the existing method. However, the performance of "$\mathcal{A}$ w. neg.entr" seems independent of $\varepsilon$ and the performance of [HHRW16] seems worse when $\varepsilon$ increases, both of which are strange. The observations here are because the optimality gap itself does not reflect the whole picture, for example, constraint violations are not reflected. We thus report constraint violations in Table 5 below. It should be clear from Table 5 that constraint violations are reduced

Table 5: Constraint violations, mean$\pm$sd. Settings are the same as in Figure 1

| | Total violation | | | Max violation | | |
|---|---|---|---|---|---|---|
| | MD_ne | MD_l2 | [HHRW16] | MD_ne | MD_l2 | [HHRW16] |
| $\varepsilon = 1$ | 7.9$\pm$1.3 | 6.7$\pm$1.7 | **4.9**$\pm$0.7 | 2.7$\pm$0.4 | 2.8$\pm$0.6 | **2.0**$\pm$0.3 |
| $\varepsilon = 2$ | 7.0$\pm$1.2 | 6.7$\pm$1.8 | **4.6**$\pm$0.7 | 2.4$\pm$0.5 | 2.7$\pm$0.7 | **1.9**$\pm$0.4 |
| $\varepsilon = 5$ | 6.4$\pm$1.0 | 5.6$\pm$1.5 | **4.3**$\pm$0.9 | 2.1$\pm$0.4 | 2.4$\pm$0.6 | **2.0**$\pm$0.4 |
| $\varepsilon = 10$ | 5.1$\pm$1.2 | 4.1$\pm$1.5 | **3.4**$\pm$0.8 | 1.8$\pm$0.4 | 1.7$\pm$0.5 | **1.6**$\pm$0.4 |
| $\varepsilon = 20$ | 3.5$\pm$1.2 | 2.9$\pm$1.0 | **2.1**$\pm$0.8 | 1.4$\pm$0.4 | 1.2$\pm$0.4 | **1.2**$\pm$0.4 |

when $\varepsilon$ increases, which indicates that our algorithm can maintain a good optimality performance while better reducing constraint violations. While [HHRW16] has the lowest constraint violations, our algorithms' constraint violations are only slightly worse.

**Runtime**    For this workforce scheduling problem, the runtime of our algorithm varies from potential function to potential function, see Table 6.

Table 6: Mean runtime (in seconds) per thousand iterations

| | MD_ne | MD_l2 | [HHRW16] |
|---|---|---|---|
| $\varepsilon = 1$ | 19.6 | 2.2 | 1.0 |
| $\varepsilon = 2$ | 19.9 | 2.2 | 1.1 |
| $\varepsilon = 5$ | 19.7 | 2.1 | 1.1 |
| $\varepsilon = 10$ | 20.6 | 2.1 | 1.0 |
| $\varepsilon = 20$ | 21.2 | 2.2 | 1.1 |

**Impact of "K_constant"**    From our experiments, we found that the constant factor of $K$ significantly impacts algorithms' performance. To avoid confusion, we want to point out that the algorithm by [HHRW16] set K_constant to be 2, a fixed value. But for all results in Appendix here, we let their K_constant change as well, in order to gain more understandings of its impact. We find that, when the constant is close to 1, the final allocations are more likely to achieve a smaller optimality gap, but violate constraints more severely. On the other side, when the constant increases, the final allocations will violate fewer constraints, but end with a larger optimality gap. In other words, the constant of $K$ trade-offs between optimality and constraint violations. To see the trade-off more clearly, we draw Figure 3. The x-axis represents the constant of $K$, the left y-axis is optimality gap, and the right y-axis is total constraint violation; and for all curves, the lower the better. It is evident that when K_constant is small, we get smaller gaps but higher constraint violations. Additionally, when $\varepsilon$ takes large values (say, $\varepsilon = 5, 10$), constraint violations of all algorithm are at almost the same levels, but our algorithms have better optimality performance. One may notice that when K_constant=1, the optimality gap is negative, which means $\mathsf{F}(\boldsymbol{x}^{\mathcal{A}}) \geq \mathsf{F}(\boldsymbol{x}^*)$. Considering $\boldsymbol{x}^{\mathcal{A}}$ may violate some constraints, this phenomenon is possible.

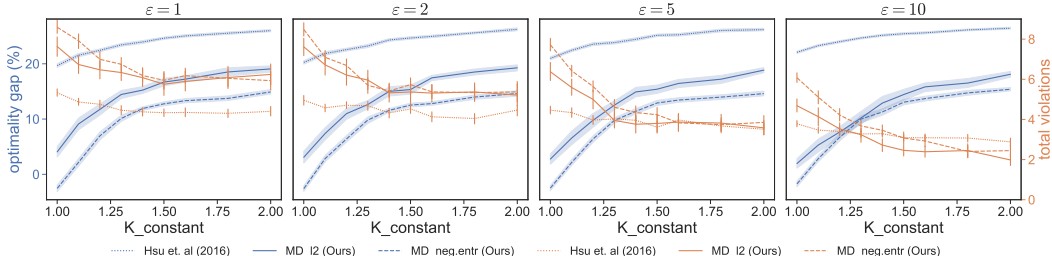

Figure 3: K_constant v.s. optimality & constraint violations. Settings are the same as in Figure 1 except K_constant. Shadow areas and error bars indicate 95% confidence interval.

## C.2 Assignment Problems

We repeat Table 3 below for discussion convenience. When reading the table, we should keep in mind

Table 7: Algorithm performance, mean±sd (this table is a copy of Table 3)

| $n$ | Algo. | Optimality gap $\frac{F(\boldsymbol{x}^*)-F(\boldsymbol{x}^{\mathcal{A}})}{F(\boldsymbol{x}^*)} \times 100\%$ | | | | Total constraint violation $\left\|\boldsymbol{v}^{\mathcal{A}}\right\|_1$ | | | |
|---|---|---|---|---|---|---|---|---|---|
| | | $\varepsilon = 1$ | 2 | 5 | 10 | $\varepsilon = 1$ | 2 | 5 | 10 |
| 800 | [HHRW16] | **-1.8**±2.6 | 4.3±3.2 | 5.3±2.8 | 3.8±1.5 | 84.8±18.4 | 46.2±16.8 | 19.0±9.3 | 11.0±5.2 |
| | MD_l2 (ours) | 1.8±3.9 | **1.3**±2.3 | 0.8±0.8 | 0.5±0.5 | 27.7±12.0 | 12.7±8.3 | 5.1±2.6 | 2.7±1.6 |
| | MD_ne (ours) | 2.1±4.0 | 2.0±2.2 | **0.7**±0.9 | **0.4**±0.5 | **23.9**±13.3 | **9.2**±6.7 | **4.7**±3.4 | **2.6**±1.5 |
| 1500 | [HHRW16] | **-6.9**±1.5 | **1.8**±1.4 | 13.8±1.3 | 12.3±0.9 | 256.7±16.0 | 156.8±20.8 | 30.3±10.8 | 15.7±6.7 |
| | MD_l2 (ours) | 4.3±3.4 | 2.8±2.3 | 1.2±0.8 | 0.7±0.7 | 54.6±18.5 | 35.1±14.8 | **12.6**±4.6 | 7.2±3.3 |
| | MD_ne (ours) | 6.0±4.1 | 3.0±2.1 | **1.1**±0.9 | **0.6**±0.6 | **47.8**±23.0 | **25.8**±12.0 | 13.4±5.9 | **4.5**±2.5 |
| 3000 | [HHRW16] | **-29.4**±0.9 | **-18.9**±1.2 | 6.9 ±1.2 | 8.2±1.0 | $> 10^3$ | $> 10^3$ | 743.8±25.7 | 680.1±29.4 |
| | MD_l2 (ours) | 4.3±5.1 | 3.0±2.4 | **2.0**±1.3 | 1.5±0.7 | 193.9±54.1 | 97.2±23.9 | 39.8±12.6 | 22.0±6.3 |
| | MD_ne (ours) | 10.4±5.2 | 6.1±3.0 | 3.0±1.1 | 1.6±0.6 | **134.5**±56.4 | **65.4**±25.1 | **27.9**±8.4 | **18.1**±6.0 |

*Notes.* Algorithm parameters: $K = 1.1\overline{u}/(\gamma\underline{b})$, $\delta = .01$, $T = 10^4$. Other settings follow Theorem 3.10. MD_l2 and MD_ne means potential function is squared $\ell_2$ and negative entropy, respectively. For three cases $n = 800, 1500, 3000$, we set resource level $\gamma = 0.1, 0.05, 0.02$, respectively. For all values in the table, the lower the better. **Bold**=better.

that optimality gaps should be understood together with constraint violations. Some observations from the table are:

1. When $\varepsilon$ is small ($\varepsilon = 1, 2$), while [HHRW16] achieves the lowest optimality gaps in most instances, it achieves them at severe constraint violations, which might be not acceptable in practice. In contrast, our algorithms have reasonably satisfying optimality performances and much smaller constraint violations.

2. When $\varepsilon$ gradually grows, our algorithms have a clear pattern of convergence, i.e., smaller optimality gap and fewer constraint violations; but the pattern for [HHRW16] is not clear enough. We conjecture that this is because of high-variance noises injected.

3. If constraint violation matters, MD_ne is most preferred.

**Runtime** Runtimes are reported in Table 8. Compared to existing methods, our algorithms do not observe a significant increase in runtime. Moreover, as we set $T = 10^4$, we can in fact complete training within one hour, even for the largest case with $n = 3000$.

**Dual convergence** Figure 4 shows the progress of dual variables convergence. The y-axis is the gap between dual variables by our algorithms and the optimal dual variables, while x-axis is the training progress. Shadow areas indicate standard deviations. It is evident that when $\varepsilon$ is larger, dual variables converge faster to optimal values, and stay around the optimal values in remaining iterations. When $\varepsilon$ is small, say $\varepsilon = 1$, dual variables still show a converging tendency. Moreover, dual variables by MD_ne converge much faster, which further justifies its better performance observed earlier.

Table 8: Runtimes per thousand iterations, in seconds

| $n$ | Algo. | $\varepsilon = 1$ | 2 | 5 | 10 |
|---|---|---|---|---|---|
| 800 | [HHRW16] | 47.0 | 49.1 | 49.3 | 48.9 |
| | MD_l2 (ours) | 49.2 | 50.6 | 48.7 | 50.0 |
| | MD_ne (ours) | 52.9 | 54.5 | 56.0 | 52.4 |
| 1500 | [HHRW16] | 161.1 | 163.5 | 158.1 | 171.3 |
| | MD_l2 (ours) | 236.0 | 242.6 | 242.3 | 228.1 |
| | MD_ne (ours) | 248.1 | 227.3 | 250.8 | 249.9 |
| 3000 | [HHRW16] | 321.3 | 348.1 | 368.9 | 379.3 |
| | MD_l2 (ours) | 345.6 | 312.9 | 333.6 | 321.0 |
| | MD_ne (ours) | 335.7 | 331.8 | 326.4 | 326.7 |

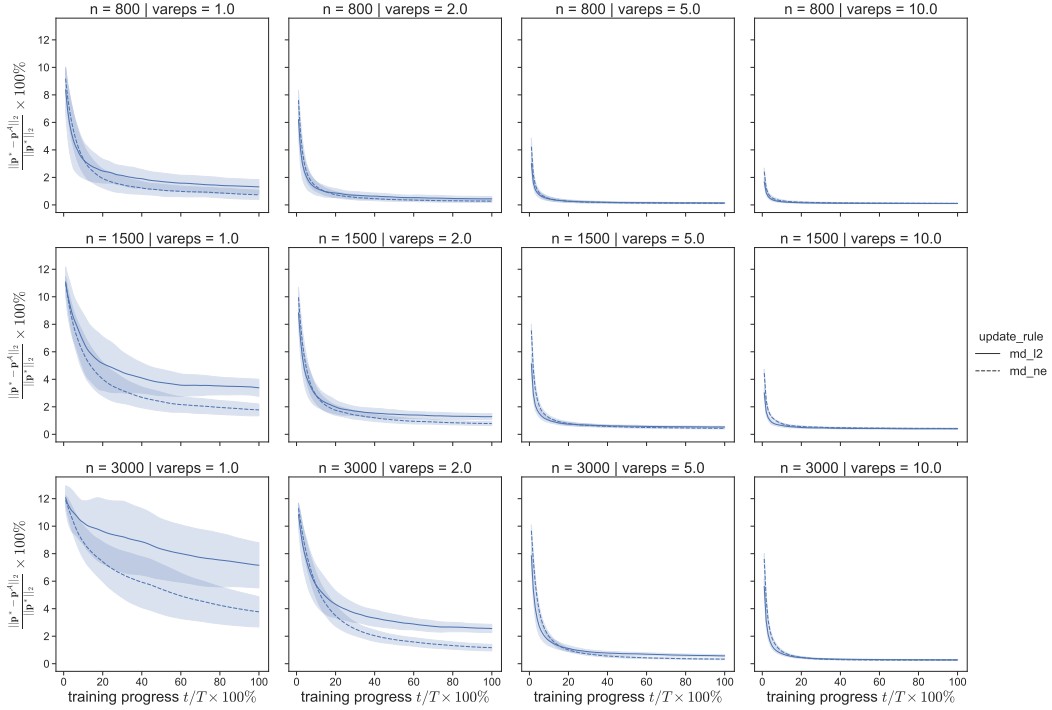

Figure 4: (prefix averaging) dual variables converge.

