# OpenReview forum: "Noisy Dual Mirror Descent: A Near Optimal Algorithm for Jointly-DP Convex Resource Allocation"
_NeurIPS.cc/2024/Conference — NeurIPS 2024 poster_

### Official Review · Reviewer_YdYP · 2024-07-16

**Soundness:** 3
**Presentation:** 3
**Contribution:** 3
**Rating:** 6
**Confidence:** 3

**Summary:**

The paper studies a class of convex resource allocation problems, in which the utilities and constraints are private (and bounded).
The paper proposes a simple algorithm that applies mirror descent to the dual problem (while the update of the primal variables is assumed to be exact).
The main technical result is an improvement of the utility bound (improves dependency on number of constraints)

**Strengths:**

- The paper is well-written, and the technical results are interesting  (though I did not carefully review proofs)
- This class of convex problems (with private utilities and constraints) is well motivated and seems relevant in practice.
- There is a concrete improvement by a factor of m in the utility upper bound for this class, under strong duality assumptions.

**Weaknesses:**

- Algorithm 1 assumes that the maximization problem (line 3) is solved exactly. This might be realistic for a subclass of problems (for instance when utilty and constraints are linear -- and I suppose this is true in experiments), but the analysis considers a much more general class, for which exact updates are generally not possible (so I find Table 1 to be a bit misleading about this). For the analysis to be complete, this would require allowing inexact updates in line 3. Can the authors work out the details in this case (please explain in the rebuttal)? Otherwise this should be clearly stated as an assumption and reflected in Table 1.
- Near optimality: The lower bound assumes that the solution is feasible, while the utility upper bound is for infeasible solutions. I don't think you can reasonably conclude that "noisy mirror descent is near optimal" (line 258). It should be made clear that optimality is still very much open. The title should be revised too.
- Dependence on $\underline{b}$ in Theorem 3.10: it seems one is trading-off a $\sqrt m$ utility improvement for a $1/\underline b^2$ term (both in utility and feasibility) which could be arbitrarily bad depending on the problem instance. Is this unavoidable? Can the authors comment?

Minor:
- I am not sure how meaningful the "compulsory requests" relaxation is, as this requires some restriction on $p^*$, and comes at the expense of simplicity/presentation.
- the introduction mentions personalized recommendation as a motivating example. Can the authors provide more details or references? How is this problem modeled as a constrained resource allocation problem?
- Theorem 3.5 and 3.6 are written in terms of $p^{(1)}$. Presumably the bounds in Table 1 use a specific choice of $p^{(1)}$. It would be good to comment on that.
- Using $u$ both for utility functions and "violation levels" (line 183) is confusing. Please choose a different notation.
- In Condition 3.1, I believe you technically want to require differentiability on relint W, and not on W (otherwise your negative entropy examples would not satisfy this -- please comment on whether this is problematic in your proofs).
- Some quantities are not defined (even if standard, please make sure to define them to make the paper self-contained), for example the $-i$ notation and strong convexity wrt $\|\cdot\|_p$.
- Assumption 2.4.3 should be rephrased. Maybe "the constant $\gamma$ in (4) is assumed to be ..."

**Questions:**

Please see weaknesses section.

**Limitations:**

Additional limitations should be discussed:
- The near optimality claim is not fully justified (given the feasibility assumption in the lower bound).
- The algorithm and analysis assume primal updates can be done exactly.

---

> ### Author Rebuttal · Authors · 2024-08-03
>
> Dear Reviewer YdYP,
>
> Thank you very much for your thorough reading and for providing insightful comments. We will respond to your concerns below one by one. We are more than happy to answer follow-up questions.
>
> **Weaknesses**
>
> - **Response:** Thank you for this sharp observation. Yes, you are right that we have to assume the maximization problem (line 3) is solved exactly. The exact solution is necessary for invoking Danskin's Theorem, which indicates that the vector $\bf{g}^{(t)}$ defined in line 5 is a gradient of the dual problem at $p^{(t)}$. The experiments only examine linear problems, which has closed-form solutions, i.e., exact solution can be obtained; so no issues for the experiments. For more general problems considered when developing theory, we believe it is not a big issue neither. Because the maximization problem is a concave problem (note that both $F(\cdot)$ and $-a(\cdot)$ are concave), it can be solved to $\epsilon$-optimality, $\epsilon\rightarrow 0$. That says, in a limiting sense, the maximization problem can always be solved exactly. Then, we can invoke Danskin's Theorem for gradients, and all proofs remain valid. However, if no exact solutions, then Danskin's Theorem is no longer valid, and the vector $\bf{g}^{(t)}$ defined in line 5 might not be a (sub)gradient; and all proofs become flawed. We think it needs an overhaul in the proof for this case, and therefore may leave it for future study. Considering that, we will highlight "solve exactly" as an assumption in Table 1. Thank you very much for pointing out this.
>
> - **Response:** Thanks. Other reviewers raised similar concerns, so we launched a response to all reviewers on this concern, please refer to that for more details.  Before talking about "optimality", we need to clarify the way we evaluate algorithm's performance: we actually treat the sum of utility bounds and constraint violations as the ultimate upper bounds. In other words, when reading Table 1, we should look at the last two columns together. Constraint violation means resources are in shortage; but purchasing more resources from an emergency supplier costs extra money. So practically, the suboptimality in utility + money spent for getting extra resources should be the ultimate performance measure.
>
>     We fully understand your concern, and we also believe an algorithm without violation should be the true optimal algorithm. So we decide to clarify our interpretation of optimality in the abstract, after Table 1 and after Theorem 3.10, 4.3. Thank you again for this valuable comment.
>
> - **Response:** Thank you for another sharp observation. First of all, we found the term $1/\underline{b}^2$ can be improved to $1/\underline{b}$. This has been updated to the revised manuscript. Second, there is indeed a trade-off between an improvement of $\sqrt{m}$ and an additional factor $1/\underline{b}$. To our best knowledge, the $1/\underline{b}$ term seems unavoidable. This term has been consistently observed in (non-private) resource allocation literature that involves analysis through primal-dual relationships, see [1-3]. Some papers interpret $\underline{b}$ as a sign of unbalance within resources, and argue that if $\underline{b}$ is very small, then no algorithm can achieve good performance [1]. In our specific case, the term $1/\underline{b}$ appears around line 509 when bounding $p^*$ under strong duality, which is finally carried to utility and feasibility bounds. So it seems inevitable. However, because $\bf{b}$ appears in the r.h.s of constraints, in practice, $\underline{b}$ is unlikely to be an arbitrarily small value (though possible in theory). The references [1-3] we provide below consistently observe a satisfying performance in experiments, despite the hidden drawbacks you raised. Ultimately, whether the trade-off matters depends on what assumptions we make on $\bf{b}$. For us, we treat it as a constant. Hope our response suffices.
>
> **Minors**
>
> - **Response:** In terms of modeling, it gives more flexibility; for example, physically-challenged students **must** be assigned an accessible seat. The condition just ensures normal students do not take away the seat from them. Technically, it facilitates the proof for lower bounds: the constructed datasets contain compulsory requests. Ultimately, this is a 'relaxation', so we think it should be fine.
>
> - **Response:** Thank you. You may find examples and references in [4] helpful. Some network revenue management problems involving personalized recommendations and limited inventory can be modeled as resource allocation problems.
>
> - **Response:** Thanks for alerting us of this. We indeed presume specific $p^{(1)}$ for Table 1. We also notice that the connection between Theorem 3.5, 3.6, 3.10 and Table 1 is somewhat weak and vague. To enhance the connection, we will add the table in the attached PDF file to our main text. The table illustrates what $p^{(1)}$ is used for what bounds.
>
> - **Response:** Thanks. We now use $\bf{v}$ to denote violation levels.
>
> - **Response:** Thanks. Yes, we want differentiability in the interior of $\mathcal{W}$. The modification does not cause problems in our proofs. Because proofs are built on results from mirror descent literature. With differentiability in int($\mathcal{W}$), our assumptions align with those for mirror descent.
>
> - **Response:** Thanks for your thorough reading. These notations are properly introduced now.
>
> - **Response:** Thank you again for your thorough reading. The poor presentation of Assumption 2.4.3 is a result of formatting the paper to meet the 9-page limit requirement. With one more page allowed now, we have rephrased this.
>
>
> [1] A dynamic near-optimal algorithm for online linear programming. OR, 2014
>
> [2] Dual mirror descent for online allocation problems, ICML, 2020
>
> [3] Online linear programming: Dual convergence, new algorithms, and regret bounds, OR, 2022
>
> [4] An improved analysis of LP-based control for revenue management, OR, 2024

---

### Official Review · Reviewer_8BYy · 2024-07-27

**Soundness:** 3
**Presentation:** 3
**Contribution:** 3
**Rating:** 7
**Confidence:** 3

**Summary:**

The paper addresses the allocation problem under Joint Differential Privacy within the framework of a convex consumption function, a concave utility function, and a convex 'personal' domain.

The contributions of this work are threefold. Firstly, it derives results similar to those in previous research, but notably, it does so using only weak duality. Secondly, the paper improves the previously established optimality gap by leveraging strong duality. Lastly, it establishes a lower bound for the Joint Differential Privacy allocation problem, specifically for algorithms whose outputs are feasible allocations, within a certain regime of the epsilon parameter.

**Strengths:**

The work effectively distinguishes between the capabilities of weak and strong duality. In the case of strong duality, it achieves a significant improvement by reducing the dependency on the quantity of resources $m$ by a square root factor. This improvement, however, comes with the introduction of polylogarithmic dependencies in both the duality gap and the constraint violation.
The authors also present a lower bound for the Joint Differential Privacy (JDP) allocation problem under the assumption of strong duality.

**Weaknesses:**

The text could benefit from increased clarity in certain areas. For example, it is not immediately clear in the abstract that the lower bound is derived using the strong duality assumption. While this is clarified later in the paper, it would be beneficial for the abstract to be more specific on this point.
Additionally, it is not clear to me in what sense the lower bound is near matching the upper bound. It seems to me that there is an extra $m$ factor in the lower bound, which makes the difference not merely polylog. Am I missing something?

**Questions:**

It caught my attention that the upper bound is for an algorithm that can violate the constraints, but the lower bound is for one that always output feasible allocations. Is it significantly more difficult to analyze a lower bound for the first type of algorithm (fixing a range of constraint violation)?

**Limitations:**

Yes, the authors have adressed the limitations of their work.

---

> ### Author Rebuttal · Authors · 2024-08-03
>
> Dear Reviewer 8BYy,
>
> We want to express our gratitude for your time in reviewing our work and the valuable comment you sent to us. Below, we reply to your concerns one by one. Feel free to let us know any follow-up questions.
>
> **Weaknesses**
>
> **Response:** Thank you very much for the suggestions regarding readability. We fully agree with you and have revised our paper accordingly. Now, the abstract clearly indicates the lower bound is derived under strong duality. As for how bounds match, we need to clarify what is the upper bound first. We basically treat the (perhaps weighted) sum of the optimality gap and constraint violations as the overall upper bound. This is reasonable as optimality gap bounds are achieved by solutions that violate constraints.
>
> Intuitively, constraints being violated means more resources are allocated to agents, and it makes sense in this situation, the utility $F(x^\mathcal{A})$ is closer to the optimal total utility $F(x^*)$ from below. In an extreme case, if $x^{\mathcal{A}}$ violated constraints too much, the value $F(x^\mathcal{A})$ might even be greater than the optimal $F(x^*)$ whose $x^*$ is feasible. The example here suggests that when evaluating algorithms that may violate constraints, we must take into account constraint violations. So when reading upper bounds in Table 1, we'd better consider optimality UB and constraint violations together, taking the sum of the two as the ultimate upper bound. If bounds are understood in this way, then our lower bound matches upper bounds up to log factors. Hope this addresses your concern.
>
> Nevertheless, we found that our current presentation fails to highlight this interpretation. To make things clear, we decided to clarify this with a new paragraph, **Our interpretation of algorithm performance**; you may find the content of this paragraph in our response to all reviewers.
>
> Thank you very much for the comments. And we are very happy to take further questions.
>
>
> **Questions**
>
> **Response:** With our preceding response, we hope the raised concern in your first sentence is well-addressed. Therefore, we move to the second sentence "is it significantly more difficult to...?" The short answer is "No", it is not significantly more difficult to analyze the lower bound for algorithms given a fixed value of constraint violation. The fixed value of constraint violation can be treated as more resources, then the current proof can still go through after slight modifications. Specifically, the constructed datasets $\mathcal{D}_0$ and $\mathcal{D}_1$ in the proof of Theorem 4.3 should now be modified to contain more compulsory requests that consume the additional resources allowed from violations. Remaining proofs then follow. But the valid range of $\varepsilon$ may need to adjust accordingly, as indicated in step 4 of our proof.
>
> Overall, whether to investigate the lower bounds of an algorithm outputting feasible or infeasible solutions still depends on the way by which we assess algorithms for constrained problems. If constraint violation is allowed, we must take violations into account and give a proper interpretation. For upper bounds, interpreting violations as additional penalties makes sense, in our opinion. But for lower bounds, we cannot come up with a reasonable explanation so we only focused on feasible algorithms.
>
> In sum, thank you very much for your valuable comments and suggestions. Hope our responses address your concerns, and we are willing to take follow-up questions, if any.

---

> > ### Comment · Reviewer_8BYy · 2024-08-14
> > **Answer to Rebuttal**
> >
> > I want to thank the authors for answering my questions. I find them satisfactory and it helped me understand the performance of their algorithm much better.

---

### Official Review · Reviewer_kLe8 · 2024-07-27

**Soundness:** 3
**Presentation:** 3
**Contribution:** 3
**Rating:** 7
**Confidence:** 4

**Summary:**

The submission studies jointly differentially private algorithms for resource allocation problems, which are a broad generalization of packing linear programs. The work addresses this challenge by considering a primal-dual formulation of the problem, and running a noisy mirror descent algorithm on the dual. Standard analyses of the mirror descent method and the impact of noise addition then provide bounds on the suboptimality and approximate feasibility of the method. The most interesting result is that under strong duality, the approach yields bounds which are polylogarithmic on the number of resources. The intuition behind this result is that strong duality provides bounds on the magnitude of an optimal dual solution. This suffices to more accurately locate the dual fesible region, which allows the use of entropy regularization for the noisy mirror descent method. The paper also provides lower bounds, which yield some insight on the possible near optimality of the method.

**Strengths:**

1. The generality of the class of allocation problems considered.
2. The insights derived from duality and the mirror descent algorithm.
3. Numerical results are encouraging.

**Weaknesses:**

1. I am confused about how the lower bound of Theorem 4.3 compares to the upper bounds. The upper bounds of the paper provide in-expectation guarantees for both suboptimality and infeasibility; moreover, the latter is quantified in an $\ell_{\infty}$-sense (which I think it's the right approach in this case). By contrast, the lower bound is expressed for algorithms which are almost surely feasible, leading to the expression that only involves the objective. Since the upper bounds do not contain linear factors in m, I don't see how this lower bound is optimal (or that it gives insight on the efficiency of the upper bound).

2. I also left some more specific comments in the questions section. I believe clarifying these it is very important for the submission to be publishable (and I also think these should be easy fixes).

**Questions:**

MAJOR QUESTIONS:
1. About point 1 in weaknesses, it would be more consistent if the authors try to:
a) Provide high-probability results for the constraint violation upper bounds (this should be easy, using the high probability tail bounds of Gaussian noise addition).
b) Provide lower bounds for joint DP resource allocation problems.
2. About the upper bounds without strong duality, it is unclear to me how the set $\mathcal{W}$ should be chosen in this case. I think that under quadratic potential they can just let $\mathcal{W}$ to be the whole space, but please clarify.


MINOR QUESTIONS:
1. In page 3, I don't understand why is it claimed that the dual may not be convex. The dual objective is written as a maximum of functions which are affine in p, so it should always be convex.
2. The paper does not accurately represent the work done on differentially private versions of mirror descent. The authors should make a more thorough search about this topic, in order to put their work in context.
3. If I understand correctly, the vector $b$ represents the resource capacities, which it is reasonable to expect them to be public information. Either way, I believe the comment right before theorem 3.10 comes way too late, and the discussion about what is private and what is public it should be brought up much earlier.
4. The paragraph right after Theorem 3.10 about "purchasing resources" I also found it very confusing. Please make efforts to clarify it.
5. The last comment at the proof ot Theorem 3.6 should not be in the appendix (neither in the proof). This is a more practical consideration that does not belong to the proof, and it should be in the main file instead.

---

> ### Author Rebuttal · Authors · 2024-08-03
>
> Dear Reviewer kLe8,
>
> We sincerely thank you for reviewing our work and for your valuable feedback. Below, we respond to the concerns you raised one by one. Please feel free to let us know if you have any further follow-up questions. We are more than happy to take them.
>
> **Weaknesses**
> 1. **Response:** Thanks for the comment. We want to first express our sincere apology for a typo in the lower bound (Theorem 4.3): the bound is supposed to be stated for an in-expectation gap $F(x^*) - E_\mathcal{A}[F(x^{\mathcal{A}})]$. But somehow we missed out the expectation symbol in the main text, though we indeed did the proof for in-expectation guarantee (see line 605). We feel sorry for any confusion caused, and the typo has been fixed.
>
>     Regarding lower and upper bounds. When reading upper bounds, we should read optimality gap bounds and constraint violation bounds together, because the optimality gap bound itself "_does not reflect the whole picture_" (line 180). As we discussed in our response to all reviewers, we therefore treat the (perhaps, weighted) sum of both gaps, i.e., optimality gap + constraint violations, as the ultimate upper bounds in our mind. The idea here follows our "purchasing resources" statement you mentioned in your next comment, and it admits a social welfare interpretation: the loss in total social welfare is the sum of (i) loss in total utility of agents and (ii) the decision maker's expenditure on extra resources.
>
>     Nevertheless, we agree with you and think our current presentation fails to make this interpretation clear. We thus decided to bring this interpretation up to the Introduction as a new paragraph **Our interpretation of algorithm performance**. The content of this new paragraph can be found in our response to all reviewers above. We sincerely thank you again for this valuable comment.
>
> 2. **Response:** Thank you very much. More replies coming below.
>
> **MAJOR QUESTIONS**
> 1. **Response**: We feel sorry again for missing out the expectation symbol. All our bounds are consistent in the sense that they are for in-expectation performance. As for high-probability bounds, it is an easy fix for optimality gap bounds, but we believe for the constraint violations, it is not easy. This is because the constraint violation bound is derived from a sandwich method. Both sides of the sandwich involve Gaussian random vectors. And the lower part of the sandwich appears to be harder for this purpose, because of the term $\sum_t <n^{(t)}, p^{(t)}>$ around line 481. In general, $p^{(t)}$ is not assumed to be bounded, and thus we may need to bound it first before employing any Gaussian tail bounds. However, as $p^{(t)}$ depends on all past $n$ and $p$, it is therefore hard to characterize, and may need an iteration-by-iteration check. At this moment, we don't have a clear idea of how to do this (by martingale process maybe?). But we will try our best and report any successful attempts. In contrast, our in-expectation bound takes advantage of the zero-mean of $n^{(t)}$ and independence between $n^{(t)}$ and $p^{(t)}$, so this sum term is zero, and proofs can go through.
>
> 2. **Response:** Thanks for the question. Yes, for the case without strong duality, the quadratic function works. We also found the current presentation fails to make the choice of potential functions clear (they are scattered among theorems). To make things clear, we decide to add the table in the attached PDF to our main text. The table shall clarify what potential functions and initial points should be chosen, and what the corresponding performance bounds are.
>
> **MINOR QUESTIONS**
> 1. **Response:** Yes, we concur. The dual is always convex. We have fixed this.
> 2. **Response:** Thanks for this suggestion. We have added discussion on noisy mirror descent and its applications in the Introduction, and found our work a position in the context. In short, noisy mirror descent is usually used for private stochastic convex optimization and saddle point problems, and is applied to a primal problem. In contrast, we apply noisy mirror descent to the dual problem and use it for resource allocation.
> 3. **Response:** Yes, you are right, the vector $\bf{b}$ is public info. We agree with you and added explanations in the Introduction on what information is public/private. Now, it should be clear from the Introduction that only $z_i$ is private data; other info is public. Thank you for this suggestion.
> 4. **Response:** Thanks for this suggestion. We have added a discussion to clarify this. As this is related to point 1 you mentioned in Weaknesses, we hope our response there has cleared your concern. Please feel free to let us know if we have made it clear to you. We will be more than happy to provide follow-up explanations.
> 5. **Response:** Thank you. We agree with you. The discussion now is brought to the end of Section 3.1, immediately after Theorem 3.6.

---

> > ### Comment · Reviewer_kLe8 · 2024-08-08
> > **Answer to Rebuttal**
> >
> > First, I would like to thank the authors for their clarifications and thorough work on improving the paper and the results. I generally agree with the authors' answer and proposals, and I am raising my score.
> >
> > there are still two things I find are worth commenting on:
> >
> > 1. About the efficiency measure. While I understand that there is a welfare interpretation that justifies merging the suboptimality gap and constraint violations, I think there is still value in stating results in a bi-criteria sense (i.e., separately addressing the suboptimality and $\ell_{\infty}$ constraint violation). This for various reasons:
> > a) Utilities and constraints could be in different scales (money transfers cans equalize scales, but it is still s.t. which may be specific of the application).
> > b) The polylog upper bounds look far less exciting when they are added (and some information is lost IMO).
> > All in all, my suggestion is that you keep both results.
> >
> > 2. About the high-probability constraint violations. That's interesting! Thank you for the clarification. If the only issue in this sandwiching is the term $\sum_t <n^{(t)}, p^{(t)}>$, I am possibly wrongt about this, but at this point you are already constraining the dual variables $p$ to lie on the simplex, and since they are predictable (and bounded) it should not be too difficult to upper bound this sum w.h.p. (usiing standard martingale arguments). There are other nontrivial bounds happening in this proof, so I am not sure whether this is enough. BTW, I think it's OK if you don't address this for the submission.

---

> > > ### Author Response · Authors · 2024-08-11
> > > **Thank you**
> > >
> > > Thank you very much for raising the score. And we are happy to know our response cleaned your concerns.
> > > As for the efficiency measure, we agree with you and also believe keeping both results would be more informative. So we will keep both.
> > >
> > > Last, thank you again for all your valuable suggestions throughout the review process. Thank you!

---

### Author Rebuttal · Authors · 2024-08-03

Dear Reviewers kLe8, 8BYy, YdYP,

We sincerely thank you for your time in reviewing our work and in providing valuable feedback. We would like to initiate a global response to a common concern raised by all of you:

- Given that upper bounds are for an algorithm that can violate constraints, but the lower bound is for one that always outputs feasible solutions, how does the lower bound compare to upper bounds, and why do we claim the proposed algorithm to be "near optimal"?

Before talking about "optimality", we need to clarify the way we evaluate algorithms' performance. **We essentially treat the sum of utility bounds and constraint violations as the ultimate upper bounds** in our mind. Because the algorithm may output infeasible solutions, solely looking at its utility upper bounds is not reasonable. That is to say, when reading performance bounds in Table 1, we should look at the last two columns together. Failing to do so may lead to false conclusions. To see why, please take a look at the table in the attached PDF. As indicated by the first two rows (or third and fourth rows), with different initial point $\bf{p}^{(1)}$, performances vary a lot. When $\bf{p}^{(1)}\rightarrow \bf{0}$, the utility gap bounds are even smaller, but this comes at huge constraint violations (so, we should not conclude that $\bf{p}^{(1)}\rightarrow \bf{0}$ is a better choice)

For this reason, we use the sum of utility bounds and constraint violations as the ultimate performance measure (this idea admits a social welfare interpretation, see text in the block below). In this sense, our proposed algorithm is near optimal.

(On a side note, [HHRW16]'s algorithm is not necessarily near optimal for general convex cases, because their analytic results are only valid for linear problems due to a supporting Lemma they invoked. We will clarify this in the revision.)

We found that the presentation in our initial submission failed to stress the performance measure we used. To this end, we will

- (i) Add the following paragraph to Introduction

> **Our interpretation of algorithm performance** Because algorithms considered may output infeasible solutions, when assessing their performance, we should take into account both suboptimality in utility and constraint violations, i.e., the last two columns in Table 1. We, therefore, treat the sum of them as the ultimate performance. This idea admits a social welfare interpretation: when the central decision maker (e.g., a government) desires to implement an infeasible allocation, she may purchase additional resources from an emergency supplier to make the allocation feasible. Then the total loss in social welfare is the (perhaps, weighted) sum of (i) the loss in utility of agents and (ii) the decision maker's expenditure on extra resources. Besides, Table 2 suggests that initial points affect performance significantly. If we only look at suboptimality, we may mistakenly conclude that the gap could be arbitrarily small. Following our interpretation of performance, our proposed algorithm is near optimal.

- (ii) Add the table in the attached PDF to the main text.

We hope our response addresses your concern. Below, we will reply to each one of you individually. Thank you very much again for all the feedback and comments.

Yours,

Authors of Submission 3277

---

### Decision · Program_Chairs · 2024-09-25

**Decision:**

Accept (poster)

**Comment:**

This paper studies resource allocation problems under joint differential privacy. On the one hand, it goes far beyond existing work, incorporating nonlinear utilities and nonconvex feasible sets. On the other hand, they prove that under suitable conditions --more precisely, strong duality-- a joint-DP instantiation of the mirror-descent algorithm attains utility and feasibility approximation bounds which are only poly-logarithmic on the number of resources. Lower bounds on the aggregate suboptimality + total constraint violation indicate the near optimality of the obtained results.

This work provides an interesting and novel application of mirror-descent methods in the context of private optimization. Moreover, the technical insights related to the role of strong duality seem entirely new and of independent interest. Therefore I recommend this paper to be accepted at the conference.

Some key open questions remain: some of them as the result of discussions in the rebuttal period. Authors are highly encouraged to include these discussions in the final version of the paper. Furthermore, there were some concerns about clarity of the setting and the meaning of the comparison between upper and lower bounds. Please seriously address these concerns in your final version of the paper.